# TIME-TRANSFORMER AAE: CONNECTING TEMPORAL CONVOLUTIONAL NETWORKS AND TRANSFORMER FOR TIME SERIES GENERATION

## ABSTRACT

Generating time series data is a challenging task due to the complex temporal properties of this type of data. Such temporal properties typically include local correlations as well as global dependencies. Most existing generative models have failed to effectively learn both the local and global properties of time series data. To address this open problem, we propose a novel time series generative model consisting of an adversarial autoencoder (AAE) and a newly designed architecture named 'Time-Transformer' within the decoder. We call this generative model 'Time-Transformer AAE'. The Time-Transformer first simultaneously learns local and global features in a layer-wise parallel design, combining the abilities of Temporal Convolutional Networks (TCNs) and Transformer in extracting local features and global dependencies respectively. Second, a bidirectional cross attention is proposed to provide complementary guidance across the two branches and achieve proper fusion between local and global features. Experimental results demonstrate that our model can outperform existing state-of-the-art models in most cases, especially when the data contains both global and local properties. We also show our model's ability to perform a downstream task: data augmentation to support the solution of imbalanced classification problems.

## 1 INTRODUCTION

Automatically generating realistic synthetic data assists in solving real-world problems when there is limited access to real data and manual generation is cumbersome and/or impractical. Deep generative models have shown considerable success in domains such as computer vision and natural language processing in the last decade. Numerous models have been introduced to produce synthetic images or text to address downstream tasks such as image in-painting (Pathak et al., 2016), text to image translation (Zhang et al., 2016), and automated captioning (Guo et al., 2017).

Although data generation is similarly important in the time series domain, there exist relatively few works that address this problem. This is due to the fact that the generated data is required to share a similar global distribution with the original time series data and also preserve its unique temporal properties. As such, generative models for time series data, especially those universally applicable to different types of time series data are relatively rare. Many existing works utilize Generative Adversarial Networks (GANs) (Goodfellow et al., 2014) for time series generation and most of these address the temporal challenges using Recurrent Neural Networks (RNNs) such as Long Short Term Memory (LSTM) (Esteban et al., 2017; Yoon et al., 2019; Pei et al., 2021). There are also approaches that use Variational Autoencoder (VAE) (Kingma & Welling, 2013) as the basic framework to generate time series data (Fabius & van Amersfoort, 2014; Desai et al., 2021). However, none of these works have succeeded in efficiently learning both local correlation and global interaction, which is crucial for time series processing.

Recently, Transformer based models have been successful in learning global features for different types of data including time series (Raffel et al., 2019; Dosovitskiy et al., 2020; Zerveas et al., 2021; Chen et al., 2021; 2022). On the other hand, models based on Convolutional Neural Networks (CNNs) have been shown to be better at extracting local patterns with their filters (Howard et al., 2017; Yamashita et al., 2018; Liu et al., 2019). Temporal Convolutional Networks (TCNs),

consisting of dilated convolutional layers Oord et al. (2016a), preserve the original local processing capability of CNNs, but also have an enhanced ability to learn temporal dependencies in sequential data (Lea et al., 2016; Bai et al., 2018). This makes them appropriate for time series modeling. Therefore, it is natural to combine Transformer and TCN together to learn better time series features. For example, some previous works use sequential combinations for time series tasks (Lin et al., 2019; Cao et al., 2021). However, such sequential designs do not consider the interaction between local and global features inherent in these datasets.

Motivated by the above observations and analysis, we propose a novel time series generative model named 'Time-Transformer AAE'. Specifically, we first select the Adversarial Autoencoder (AAE) Makhzani et al. (2015) as the generative framework due to its success at learning different types of tasks. The Time-Transformer is designed as part of the decoder to effectively learn and integrate both local and global features. In each Time-Transformer block, the temporal properties are learnt by both a TCN layer and a Transformer block. They are then connected through a bidirectional cross attention block to fuse local and global features. This layer-wise parallel structure along with bidirectional interaction, combines the advantages of the TCN and Transformer models: the ability of the TCN to efficiency extract local features, as well as the Transformer's ability in building global dependency. We evaluate our proposed Time-Transformer AAE on different types of time series data including artificial and real-world datasets. Experiments show that the proposed model surpasses the existing state-of-the-art (SOTA) models in addressing the time series generation task. Furthermore, we also show our model's effectiveness on a downstream task - imbalanced classification - using several real-world datasets. To summarize, our contributions are as follows:

- We propose a new time series generative model called Time-Transformer AAE, which effectively combines the advantages of TCN and Transformer in extracting local and global patterns respectively.

- We introduce the Time-Transformer module that simultaneously learns local and global features in a layer-wise parallel design and facilitates interaction between these two types of features by performing feature fusion in a bidirectional manner.

- We show empirically that the proposed Time-Transformer AAE can generate better synthetic time series data, with respect to different benchmarks, compared to SOTA methods.

## 2 RELATED WORKS

### 2.1 TIME SERIES GENERATION

Deep generative models (DGMs) have gained increasing attention since their introduction. Kingma & Welling (2013) propose a Variational autoencoder (VAE) that uses Bayesian method to learn latent representations and turn the classic autoencoder into a generative model. Goodfellow et al. (2014) introduce an adversarial approach to shape the output distribution and propose the Generative adversarial networks (GANs). Makhzani et al. (2015) combine the previous two models together in Adversarial autoencoders (AAE). They use the adversarial training procedure to perform variational inference in the VAE. Numerous models have been designed based on these basic generative frameworks and shown superior performance in image and text processing (Oord et al., 2016c;b; Pathak et al., 2016; Zhang et al., 2016; Karras et al., 2017; Arjovsky et al., 2017; Guo et al., 2017; Kadurin et al., 2017; He et al., 2019; Ahamad, 2019).

Successes in the fields of graphs and text have led to the application of DGMs in the time series domain. Most of them are derived from the GAN framework with additional modifications to incorporate temporal properties. The first of these, called C-RNN-GAN (Mogren, 2016), directly uses the GAN structure with LSTM to generate music data. Esteban et al. (2017) propose a Recurrent Conditional GAN (RCGAN) which uses a basic RNN as generator and discriminator and auxiliary label information as condition to generate medical time series. Since then, a number of works have utilized similar designs to generate time series data in various fields including finance, medicine and the internet (Zhou et al., 2018; Hartmann et al., 2018; Chen & Jiang, 2018; Koochali et al., 2019; Wiese et al., 2020; Smith & Smith, 2020; Lin et al., 2020). TimeGAN Yoon et al. (2019) introduces embedding function and supervised loss to the original GAN framework to generate universal time series. Pei et al. (2021) proposes RTSGAN based on WGAN (Arjovsky et al., 2017) and autoencoder. It focuses on real-world data generation and achieves good performance. Jeha

et al. (2022) utilizes the progressively growing architecture introduced by Karras et al. (2017) to generate long time series. In addition, VAE has also been used for time series generation. Fabius & van Amersfoort (2014) designed a recurrent VAE to synthesize time series data. The recently introduced TimeVAE Desai et al. (2021) implements an intepretable temporal structure together with VAE, and achieves state-of-the-art results on universal time series generation. Jarrett et al. (2021) uses contrastive learning framework in stead of traditional DGMs to generate universal time series and achieves good results.

In contrast to sampling from a learnt distribution, another line of works focus on step-wise generation. They borrow the idea from the variational inference of VAE Kingma & Welling (2013) and build generative models to model the sequential data at each time step so that the future steps can be inferred based on the previous ones and the ground truth data. Some early works modify the basic RNN using the variational inference and get good results with regard to log-likelihood Chung et al. (2015); Fraccaro et al. (2016). Some recent works combine this idea with TCN to produce the latest state of the art models in this area Lai et al. (2018); Aksan & Hilliges (2019). Though different to our work, these works provide an interesting direction for future time series generation.

### 2.2 TEMPORAL CONVOLUTIONAL NETWORKS AND TRANSFORMER

Temporal Convolutional Networks (TCNs) (Lea et al., 2016; Bai et al., 2018) use dilated causal convolutions in WaveNet Oord et al. (2016a) to encode temporal patterns and avoid any information leakage from future to past. TCN based models have since been used for sequential data such as time series and show the ability to successfully extract local features (Sen et al., 2019; Zhao et al., 2019; Hewage et al., 2020; Deldari et al., 2021). Transformer Vaswani et al. (2017) implements a self-attention mechanism on the entire data to learn global interactions between each point, which enhances the models' ability to learn long range dependence and improves its performance on machine translation tasks. Its variants also achieve great success on language processing, computer vision and time series tasks (Raffel et al., 2019; Zhang et al., 2019; Dosovitskiy et al., 2020; Zerveas et al., 2021; Chen et al., 2022). Some works combine these two types of models to take advantage of both their capabilities Lin et al. (2019); Cao et al. (2021). They link the two models sequentially, which assumes dependencies exist between the models. However, if the models are required to learn different levels of features separately to preserve their independence, a sequential combination is no longer suitable and instead, a parallel structure like our work is needed.

## 3 METHODS

In this section, we first describe the problem. Then, we introduce the proposed Time-Transformer AAE architecture. Afterwards, we discuss details of the model using one Time-Transformer block.

### 3.1 PROBLEM FORMULATION

Generally, multivariate time series generation via deep generative models like GANs involves training a model to learn how to map an arbitrary prior distribution $p$ to the real data distribution $p_d$, so that the model can produce realistic synthetic data $x' \sim p_d$ based on any samples $s$ drawn from $p$. For time series data, we assume each time series $x = \{t_1, t_2, \ldots, t_n\}^T \in \mathbb{R}^{n \times C}$, where $t_i = \{t_i^{(1)}, \ldots, t_i^{(C)}\}$ is the $C$ observations at time $i$ ($C$ is also called 'channel' in the rest of the paper), contains both local features and global dependencies. Thus, the map from $p$ to $p_d$ here has to represent both types of features in order to generate realistic data. The goal of our work is to design a model to learn such a map that can represent both local processing and global interaction, in order to generate realistic and useful synthetic time series data that can be used in downstream machine learning problems.

### 3.2 TIME-TRANSFORMER AAE

#### 3.2.1 OVERVIEW

We choose Adversarial Autoencoder (AAE) as our generation framework due to the potential of extending it to supervised and semi-supervised learning settings (details of AAE can be found in

Appendix A). To modify the original AAE for time series generation, we first use Convolutional Neural Networks (CNNs) as the basis of the AAE, and then insert the Time-Transformer (see Figure 1b) into the decoder of the AAE. The overall structure is shown in Figure 1a. Here, we choose

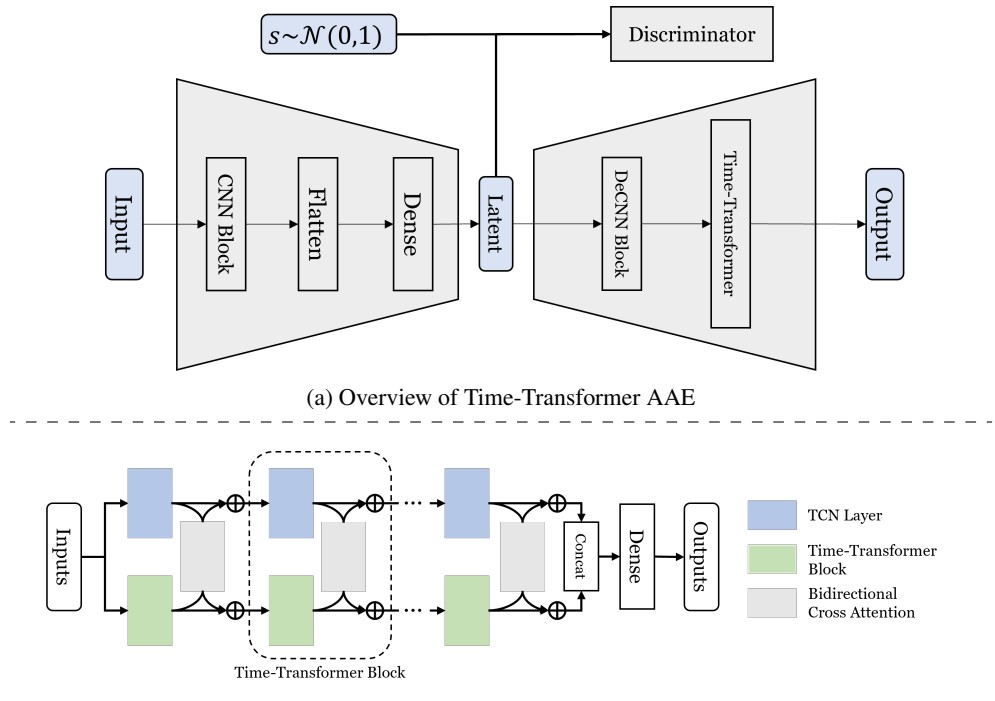

(a) Overview of Time-Transformer AAE

(b) Time-Transformer Structure

Figure 1: Time-Transformer AAE

Gaussian as the prior distribution $p(z) = \mathcal{N}(0, 1)$ and insert the Time-Transformer after a De-Convolutional block. With this design, we expect the De-Convolutional block to first reconstruct a prototype of the time series, and then the Time-Transformer to learn the detailed local/global features and generate realistic data.

Within the Time-Transformer, we use a layer-wise parallelization to combine the Temporal Convolutional Networks (TCNs) and the Transformer. The learnt prototype from the De-Convolutional block is passed into a TCN layer and a Transformer block simultaneously. The TCN learns the local features of the time series, and the Transformer finds the global patterns of the data. The learnt results then go through a cross-attention block to fuse with each other bidirectionally. At the end of this parallel structure we concatenate the outputs from both sides, and use a full-connected layer to map them into the expected dimension ($L \times C$ where $L$ and $C$ are length and number of channels of the time series respectively) and reshape into the original time series dimensions.

### 3.2.2 TIME-TRANSFORMER BLOCK

The Time-Transformer consists of several Time-Transformer blocks, which have two key differences to the standard TCN layers and Transformer blocks: (1) the layer-wise parallel design to combine local-window self-attention and depth-wise convolution and (2) the bidirectional cross attention over the two branches. Figure 2 shows the details of the Time-Transformer block.

**Layer-wise Parallelization**: In Mobile-Former (Chen et al., 2021), the authors show the advantage of using a parallel structure. They combine MobileNets (a light-weight CNN) and Transformer in parallel, and achieve better performance than sequentially combined models on image classification and object detection tasks. Inspired by this design, we also combine TCN and Transformer in a parallel manner for time series generation. However, instead of simply combining the entire TCN and Transformer, we only use one layer from the TCN and one block from the Transformer in each Time-Transformer block.

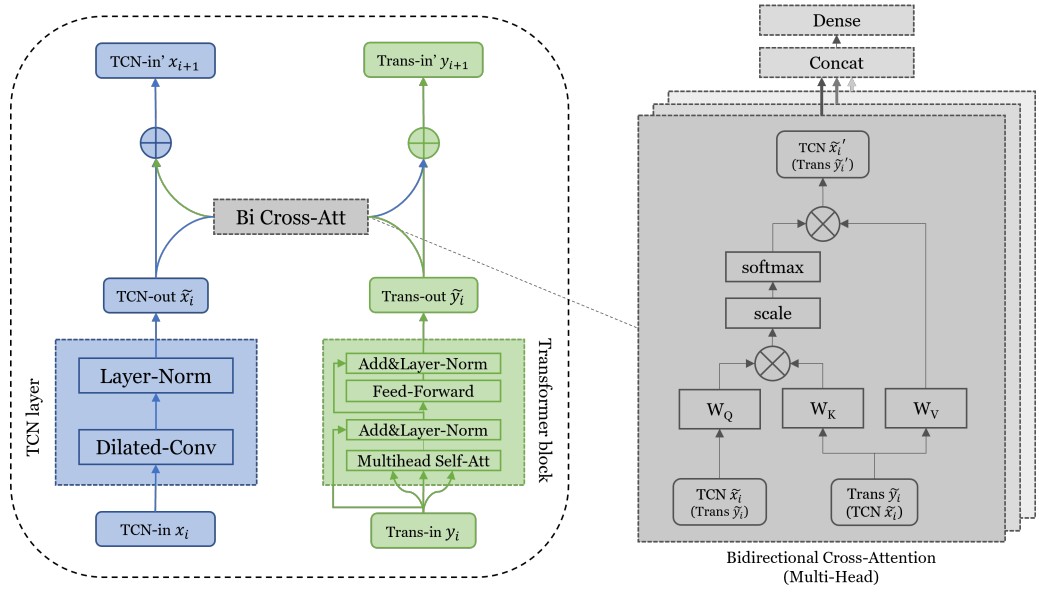

Figure 2: Detailed design of the Time-Transformer block

As shown in Figure 2, each Time-Transformer block has two streams parallel to each other, the TCN stream and the Transformer stream. Inside the TCN stream we have a dilated convolutional layer followed by a post layer normalization, which make up one hidden layer of the TCN model. The inputs and outputs of each block (TCN-in and TCN-in' respectively in Figure 2) are time series data containing local features including those learnt directly from the last/current TCN layer (TCN-out in Figure 2) and the fusion of the local and global features. On the other side, the Transformer block has the default design in accordance with (Vaswani et al., 2017): a Multi-head Self Attention layer and a Feed-forward Network, both with post layer normalization. Similarly, the inputs/outputs (Trans-in and Trans-in' respectively in Figure 2) of this side are global features from last/current Transformer block together with a feature fusion.

**Bidirectional Cross Attention**: The bidirectional cross attention block aims to build interaction between the two parallel branches, and thus fuse local and global features. As illustrated on the right hand side of Figure 2, the output of the TCN layer $\widetilde{x_i} \in \mathbb{R}^{L \times C}$ and the output of the Transformer block $\widetilde{y_i} \in \mathbb{R}^{L \times C}$ interact in a mutual manner within the cross attention block, where it bidirectionally fuses the local feature map $\widetilde{x_i}$ and the global feature map $\widetilde{y_i}$. Specifically, the TCN features are updated through a residual connection with the attention matrix to obtain $x_{i+1}$:

$$x_{i+1} = \widetilde{x_i} + \mathcal{A}_{\widetilde{y_i} \to \widetilde{x_i}} \cdot y_i W_{ev} \tag{1}$$

where $W_{ev}$ is the learnable parameter for the value embedding layer, and $\mathcal{A}_{\widetilde{y_i} \to \widetilde{x_i}}$ is the affinity matrix from Trans to TCN which can be calculated with matrix multiplication and a softmax function:

$$\mathcal{A}_{\widetilde{y_i} \to \widetilde{x_i}} = softmax(\frac{\widetilde{x_i} W_{eq} \cdot (\widetilde{y_i} W_{ek})^T}{\sqrt{C}}) \tag{2}$$

where $W_{eq}$ and $W_{ek}$ are learnable parameters of two linear projection layers. Similarly, the updated Trans feature map $y_{i+1}$ is defined as:

$$y_{i+1} = \widetilde{y_i} + \mathcal{A}_{\widetilde{x_i} \to \widetilde{y_i}} \cdot x_i W_{dv}$$
$$\mathcal{A}_{\widetilde{x_i} \to \widetilde{y_i}} = softmax(\frac{\widetilde{y_i} W_{dq} \cdot (\widetilde{x_i} W_{dk})^T}{\sqrt{C}}) \tag{3}$$

where $W_{dq}$, $W_{dk}$, and $W_{dv}$ are learnable parameters of three linear projection layers. $\mathcal{A}_{\widetilde{x_i} \to \widetilde{y_i}}$ is the affinity matrix from TCN to Trans.

## 4 Experiments

### 4.1 Datasets

We use six datasets to evaluate our model. The first three datasets have been used in several previous works (Yoon et al., 2019; Pei et al., 2021; Desai et al., 2021). Hence, they are referred to as *preliminary datasets*:

- **Sine_Sim**: A dataset containing 5000 samples of multivariate sinusoidal sequences with different frequencies $f$, amplitudes $\alpha$, and phases $\varphi$. In each dimension $d \in \{1, \ldots, 5\}$, the time series $x_t^{(d)}$ is generated using $x_t^{(d)} = \alpha sin(2\pi f t + \varphi)$ where $\alpha \sim \mathcal{U}[1, 3], f \sim \mathcal{U}[0.1, 0.15]$ and $\varphi \sim \mathcal{U}[0, 2\pi]$.

- **Stock**: 3686 samples of stock-price sequences from historical Google daily stocks. Each sample has 6 features correlated with each other, including volume and high, low, opening, closing, and adjusted closing prices. Hence, this time series data is 6-dimensional.

- **Energy**: The UCI Appliances energy prediction dataset contains time series data with high dimensionality (28) and correlated features. There are totally 19735 continuous measurements in this dataset.

The next three datasets contain time series with both local and global patterns. They are used to evaluate the model's ability to learn both types of features, and are termed *local-global datasets*:

- **Sine_Cpx**: Another 5000 samples of 5-dimensional sinusoidal sequences, which are more complex than the previous ones. They are simulated using sum of standard sinusoidal waves $x_t^{(d)} = \alpha_1 sin(2\pi f_1 t + \varphi_1) + \alpha_2 sin(2\pi f_2 t + \varphi_2) + \alpha_3 sin(2\pi f_3 t + \varphi_3)$ where $\alpha_i \sim \mathcal{U}[1, 3], f_i \sim \mathcal{U}[0.1, 0.15], \varphi_i \sim \mathcal{U}[0, 2\pi], i = 1, 2, 3$ and $\alpha_i \neq \alpha_j, f_i \neq f_j, \varphi_i \neq \varphi_j$, when $i \neq j$. Thus, in each dimension, the sequence is a sine wave globally, and also contains local patterns within each wave.

- **Music**: A waveform audio file (WAV) of the classical music 'Canon in D' (converted from an MP3 file). The file is sampled at $8000Hz$ frequency and double channels, which results in a 2845466-step, 2-dimensional time series. The music data also contains local patterns and seasonality globally. We select a 10000-step segment of the original time series, which corresponds to the original music from 2'05" to 2'06".

- **ECochG**: A medical dataset of historical patient data from cochlear implant surgeries. Each instance contains a univariate time series that represents a patient's inner ear response during the surgery. In general, each time series is a stochastic trend globally, with some local drops. The detailed background of this dataset can be found in Appendix.

### 4.2 Baseline Models

We select four previous models to compare with. The inclusion criteria are: relevance of the work, accessibility of the code, executability of the code, and performance of the model. TimeVAE and RTSGAN (Desai et al., 2021; Pei et al., 2021) are the two most recent models that have achieved state-of-the-art (SOTA) performance. TimeGAN and RCGAN (Yoon et al., 2019; Esteban et al., 2017) are two earlier methods that have gained much attention, both having achieved SOTA performance up to their introduction and have been used as a basis of comparison for many methods. All these works provide public access to their code, together with detailed instructions. We use the settings of the original designs as they have already been optimised by their authors. Details of our model's settings (hyper-parameters and data pre-processing) can be found in Appendix C. For reproducibility, we provide source code and corresponding instructions in supplementary files.

### 4.3 Benchmarks

To evaluate these generative models, we use a range of metrics and consider several desiderata for the generated time series: (1) They should be distributed close to the original data and different from each other, (2) They should be indistinguishable from the original data, and (3) They should preserve the temporal properties of the original data so that they can be used for downstream tasks.

- **Visualisation** is an indicator of (1) above as it shows the distributions over the original and generated data. We flatten the temporal dimension and use t-SNE (van der Maaten & Hinton, 2008) plots to embed them into 2-dimensional space for viualisation.

- **Fréchet Inception Distance** (FID) evaluates the quality of generated data for GAN type models. Following the ideas from (Hartmann et al., 2018; Smith & Smith, 2020; Jeha et al., 2022), we replace the Inception model of the original FID with a recently proposed state-of-the-art time series representation learning method called TS2Vec (Yue et al., 2021). We pre-train the TS2Vec on each dataset and use it to get the FID score of the model. The FID score provides a quantitative assessment of (1).

- **Discriminative Score** quantitatively measures the indistinguishability of the generated data. We follow the protocol from Yoon et al. (2019) when defining this and the next benchmarks. We first train a post-hoc sequential classifier (a 2-layer LSTM) to distinguish original and generated data. Then we get the test accuracy $acc_{te}$ from a held-out test set. Good generated time series data should be indistinguishable to the real data (accuracy close to 0.5). Thus, we use $|acc_{te} - 0.5|$ as the Discriminative Score to measure the indistinguishability (2).

- **Predictive Score** reflects how well the generated data inherit the predictive properties of the original data. We train a post-hoc sequential predictor (a 2-layer LSTM) to do one-more-step prediction, and calculate the mean absolute error (MAE). Here, we use the 'Train on Synthetic, Test on Real' (TSTR) technique introduced by Esteban et al. (2017). Good synthetic data should preserve the temporal properties of the original data, and thus the predictor should be able to learn from the generation and return good predictions of the real data. The Predictive Score provides a quantitative assessment of (3).

## 4.4 EXPERIMENTAL RESULTS

In this section, we first report the performance of the models with respect to the aforementioned benchmarks. For the three scores, we execute the experiment several times to get the averages and confidence intervals (95% confidence). Afterwards, we conduct some further studies using several real-world datasets.

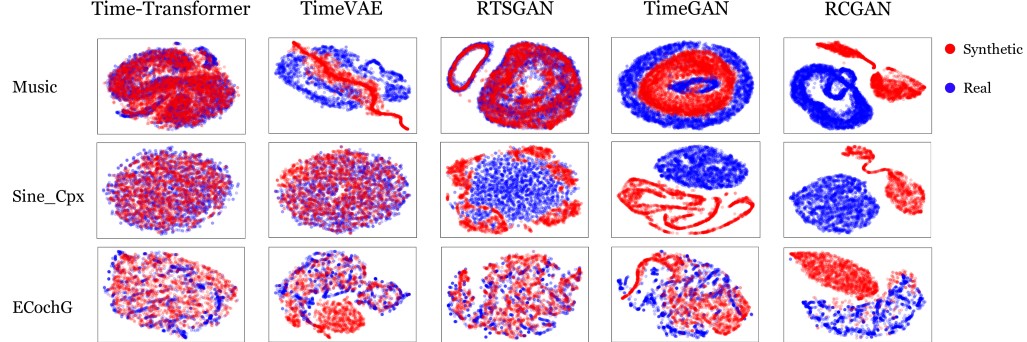

Figure 3: t-SNE visualisations of models (columns) on 3 datasets (rows). The blue dots represent original data and the red dots represent synthetic data. Heavier overlaps represent better synthesis

**Visualisation**: Figure 3 shows visualization results of three local-global datasets (results of preliminary datasets can be found in Appendix E.1). Each row contains plots of one dataset and each column represents a model. The blue and red dots represent original data points and generations respectively. It is obvious that our proposed Time-Transformer AAE (denoted as **Time-Transformer** hereafter) consistently produces synthetic data closely distributed to the original data. TimeVAE and RTSGAN also have good distributions for most datasets.

**FID Score**: Table 1 shows the FID scores of each model on each dataset. These scores are mostly consistent with the observations from Figure 3: For example, our proposed Time-Transformer obtains the best scores for all the datasets except 'Stock' where RTSGAN is slightly better. TimeVAE also generates well distributed data for this dataset, but as shown in the figure, the data points tend

Table 1: FID score: lower scores are better

| Models | Time-Transformer | TimeVAE | RTSGAN | TimeGAN | RCGAN |
|---|---|---|---|---|---|
| Sine_Sim | **0.283±0.023** | 0.766±0.027 | 3.175±0.142 | 4.572±0.153 | 9.705±0.417 |
| Stock | 0.273±0.035 | 0.303±0.037 | **0.254±0.031** | 0.645±0.081 | 22.673±0.975 |
| Energy | **0.983±0.039** | 1.598±0.067 | 6.292±0.265 | 2.821±0.117 | 20.475±0.732 |
| Music | **0.395±0.013** | 13.415±0.937 | 0.957±0.095 | 1.894±0.121 | 3.541±0.263 |
| Sine_Cpx | **1.502±0.062** | 7.696±0.251 | 3.486±0.170 | 14.255±0.491 | 18.922±0.736 |
| ECochG | **0.348±0.024** | 5.197±0.237 | 0.527±0.062 | 0.674±0.098 | 12.652±0.617 |

to fall on/around a line. This causes its FID score to be worse than the former two models. For those datasets where the models show similar visualisation results (e.g. Time-Transformer and TimeVAE on 'Sine_Sim') the FID scores provide a way to distinguish the quality of the generation.

**Discriminative Score**: As shown in Table 2, our model performs best in all datasets except 'Stock' where RTSGAN takes the leading position. The Time-Transformer dominates the performances within the local-global datasets where the generations from other models are highly distinguishable while those from our model show much better indistinguishability.

Table 2: Discriminative score: $|acc_{te} - 0.5|$, lower scores are better

| Models | Time-Transformer | TimeVAE | RTSGAN | TimeGAN | RCGAN |
|---|---|---|---|---|---|
| Sine_Sim | **0.131±0.021** | 0.217±0.015 | 0.489±0.007 | 0.485±0.008 | 0.500±0.000 |
| Stock | 0.463±0.023 | 0.476±0.050 | **0.374±0.022** | 0.486±0.030 | 0.500±0.000 |
| Energy | **0.496±0.005** | 0.499±0.002 | 0.499±0.001 | 0.500±0.000 | 0.500±0.000 |
| Music | **0.160±0.054** | 0.495±0.008 | 0.489±0.009 | 0.494±0.006 | 0.497±0.009 |
| Sine_Cpx | **0.168±0.041** | 0.471±0.016 | 0.489±0.031 | 0.500±0.000 | 0.499±0.001 |
| ECochG | **0.103±0.012** | 0.474±0.016 | 0.405±0.004 | 0.424±0.015 | 0.496±0.003 |

**Predictive Score**: To avoid long decimal numbers while highlighting significant differences, we report ten times MAE in Table 3. An additional column 'Oracle' shows results from the 'Train on Real, Test on Real', which represent the theoretical best performances. As shown in the table, our model performs best in all datasets except 'Stock' and 'Energy' where RTSGAN and TimeVAE show better performance. Note that our model's scores in local-global datasets are close to the original ones, which indicates that it has learnt most of the predictive properties of these data.

Table 3: Predictive score: 10×MAE, lower scores are better

| Model | Oracle | Time-Transformer | TimeVAE | RTSGAN | TimeGAN | RCGAN |
|---|---|---|---|---|---|---|
| Sine_Sim | 0.038±0.012 | **0.051±0.015** | 0.108±0.031 | 0.789±0.069 | 0.465±0.055 | 1.135±0.324 |
| Stock | 0.015±0.003 | 0.048±0.004 | 0.080±0.003 | **0.036±0.003** | 0.094±0.007 | 1.310±0.051 |
| Energy | 0.044±0.004 | 0.077±0.007 | **0.072±0.006** | 0.228±0.022 | 0.197±0.007 | 0.733±0.021 |
| Music | 0.014±0.008 | **0.017±0.009** | 0.077±0.051 | 0.021±0.009 | 0.027±0.007 | 0.179±0.016 |
| Sine_Cpx | 0.029±0.008 | **0.032±0.006** | 0.055±0.015 | 0.070±0.006 | 0.527±0.025 | 1.785±0.075 |
| ECochG | 0.011±0.009 | **0.013±0.006** | 0.087±0.008 | 0.176±0.021 | 0.042±0.006 | 0.402±0.170 |

**Ablation Study**: Using the 'Music' dataset, we investigate how each component of the model contributes to the final result. Specifically, we (1) use only a De-convolutional block, (2) add only a TCN block after (1), (3) add only a Transformer block after (1), (4) add a sequential combination of TCN and Transformer after (1), and (5) use our proposed Time-Transformer. The results of this study are shown in Table 4 (results of the other two local-global datasets can be found in the Appendix E.2). From the FID column we can see a clear improvement when adding either a TCN or a Transformer. The sequential combination also slightly improves the performance. However, with our proposed parallel structure, the Time-Transformer achieves an approximate 40% improvement compared to the sequential one. The other two columns also reflect similar situations. All these phenomena indicate the strong ability of the proposed Time-Transformer AAE to generate time series.

Table 4: Ablation study

| Components | FID | Discriminator | Predictor |
|---|---|---|---|
| De-Convolution | 2.298±0.284 | 0.490±0.006 | 0.037±0.002 |
| TCN | 0.672±0.097 | 0.471±0.013 | 0.026±0.007 |
| Transformer | 0.815±0.039 | 0.479±0.009 | 0.027±0.006 |
| TCN+Trans (Sequential) | 0.627±0.031 | 0.455±0.021 | 0.022±0.008 |
| Time-Transformer AAE | **0.395±0.013** | **0.160±0.054** | **0.017±0.009** |

**Model Application**: We briefly evaluate the utility of our model on imbalanced classification using the ECochG dataset and three real-world datasets from the UCR archive (Dau et al., 2018): Wafer, Herring, and SwedishLeaf. The details of these datasets can be found in Appendix F. Previous studies argued normal oversampling approaches (augmenting data via replication or simple modifications like jittering) can cause overfitting or create out-of-distribution synthesis (Yap et al., 2014; Yan et al., 2019). Our proposed model is expected to address these issues. Table 5 lists the classification results on the ECochG dataset using an off-the-shelf Multi-layer Perceptron (MLP) with (1) no augmentations, (2) augmentations from repeating minority class (3) augmentations from jittering method which augments data by adding noises, (4) augmentation from RTSGAN, (5) augmentation from TimeVAE, and (6) augmentation from our model (We do not list TimeGAN and RCGAN because training data here are mostly not enough to train their models). The results on the other three datasets can be found in Appendix E.3. We can observe from the table that one crucial problem

Table 5: Data augmentation evaluation (higher scores are better)

| Datasets | Components | Accuracy | Recall | Precision | AUC_ROC | AUC_PRC |
|---|---|---|---|---|---|---|
| | No Aug | 0.9751 | 0.7062 | 0.8706 | 0.9824 | 0.8521 |
| | Replication | 0.9719 | **0.9552** | 0.7072 | 0.9824 | 0.8636 |
| ECochG | Jittering | 0.9757 | 0.7074 | 0.8864 | 0.9866 | 0.8870 |
| | RTSGAN | 0.9702 | 0.6791 | 0.8235 | 0.9767 | 0.7352 |
| | TimeVAE | 0.9738 | 0.7388 | 0.8319 | 0.9807 | 0.7906 |
| | **Time-Transformer** | **0.9802** | 0.8178 | **0.9163** | **0.9945** | **0.9345** |

of imbalanced classification is low recall rates, which is caused by an under-represented positive class in the training set. The model tends to predict data as negative since it has seen much more negatives than positives. Repeating the minority class can improve the recall at the cost of precision, which reflects the aforementioned overfitting problem. However, augmenting the training set with synthetic data from our model shows better results with respect to all metrics suggesting that it can aid in solving imbalanced classification problems.

**Additional Experiments**: We further investigate our model's performance with respect to: different training sizes, local & global feature extraction, different encoder types, longer time series data, and different evaluation model. Due to space limitations, we provide the details of these experiments in Appendix E.4 to E.8 respectively.

## 5 CONCLUSION

In this paper we introduce a novel time series generative model called Time-Transformer AAE, which contains an adversarial autoencoder (AAE) and a key component named Time-Transformer in the decoder. Via a layer-wise parallelization and a bidirectional cross attention, Time-Transformer exploits the learning ability of Temporal Convolutional Networks and Transformer in extracting local features and building global interaction respectively. Through multiple experiments, we show the effectiveness of our proposed Time-Transformer AAE on time series generation tasks. Additionally, by training on different sizes of data, we show the proposed model can achieve competitive performance to state-of-the-art methods even with less training data. Furthermore, we see the model's utility in augmenting data for imbalanced classification problems. Possible future directions of our work can be: (1) altering the model setting with conditional generation in order to produce user-defined synthetic data and (2) extending the model to work with partial time series.

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

## A    ADVERSARIAL AUTOENCODER

The data generation in our model is based on a framework called Adversarial Autoencoder (AAE) (Makhzani et al., 2015) which is shown in Figure 4. It uses the adversarial training procedure from GANs to perform variational inference, and hence turns the decoder into a deep generative model.

Let us assume $x$ is the input data and $z$ is the corresponding latent code produced by a deep encoder $E(\cdot)$. Let $p(z)$ be an arbitrary prior distribution, $q(z|x)$ be the encoding distribution, and $p_d(x)$ be the data distribution. Then, we can define an aggregated posterior distribution $q(z)$ on the latent code vector:

$$q(z) = \int_x q(z|x)p_d(x)dx \tag{4}$$

The AAE attaches an adversarial network (a discriminator that distinguishes positive samples $s \sim p(z)$ from negative samples $z \sim q(z)$) on top of the latent code vector which guides the aggregated posterior $q(z)$ to match the prior distribution $p(z)$ (see Figure 4).

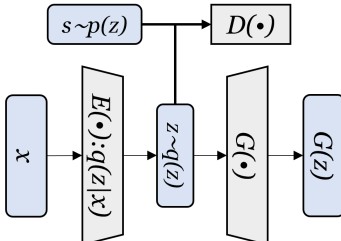

Figure 4: Adversarial Autoencoder

The objective function of the AAE consists of two parts: reconstruction loss and adversarial loss. The reconstruction loss $l_{rec}$ ensures that generator (decoder) $G(\cdot)$ can reconstruct the original data from the latent codes produced by the encoder $E(\cdot)$. Normally, it is measured using similarity $sim(\cdot)$ between the original and reconstructed data:

$$l_{rec} = \mathbb{E}_{x \sim p_d(x)} sim(x, G(z)) \tag{5}$$

The adversarial loss $l_{adv}$ provides a solution for a min-max adversarial game between a generative model $G(\cdot)$ and a discriminative model $D(\cdot)$:

$$l_{adv} = \min_G \max_D \mathbb{E}_{x \sim p_d(x)}[\log D(x)] + \mathbb{E}_{z \sim p(z)}[\log(1 - D(G(z)))] \tag{6}$$

## B    ECochG DATASET

Intra-operative electrocochleography (ECochG) is used to monitor the response of the cochlea to sound during a Cochlear Implant surgery. This surgery is a cost-effective solution for hearing impairment which, however, has been received by only a few patients due to the high probability of losing their natural hearing due to trauma caused during the surgery. Previous studies show that the changes of some ECochG components can reflect such trauma (Campbell et al., 2016; Dalbert et al., 2016). Specifically, Campbell et al. (2016) demonstrate that a 30% drop in amplitude of the 'Cochlear Microphonic' (CM - one component of the ECochG signal) leads to poorer natural hearing preservation, and thus can be used to predict trauma. This has motivated researchers to develop machine learning models to automatically detect these drops and assist the surgeon in preventing trauma during the surgery (Wijewickrema et al., 2022).

## C    EXPERIMENTAL SETUP

### C.1    DATA PRE-PROCESSING

For the preliminary datasets, we follow the settings in previous work (Yoon et al., 2019; Desai et al., 2021). We use a sliding window of 24 window size to sample the data resulting in time series of

length 24, and re-scale all data to $[0, 1]$ using the min-max normalization formula:

$$x' = \frac{x - min(x)}{max(x) - min(x)}$$

where $x$ is the original data, $min(x), max(x)$ are minimum and maximum of the data respectively.

We choose a longer length (128) for time series in local-global datasets, to preserve more local and global patterns, and to evaluate the model's ability to process longer data. We use min-max normalization to re-scale all the data to $[0, 1]$ as mentioned previously.

## C.2 MODEL SETTINGS

We search the hyper-parameters of the autoencoder based on the average reconstruction performances on the preliminary datasets. We divide the datasets into training and validation sets, and search the hyper-parameters including layer number, filter number, kernel size, stride number, head number, head size, and dilation rate using the training set. The value range of each parameter is shown in Table 6. As a result, the encoder is a 3-layer 1-dimensional CNN with 64, 128, and 256 fil-

Table 6: Value range of Hyper-parameters

| Hyper-parameters | Value range |
|---|---|
| layer number | 2,3 |
| filter number | 64,128,256,512 |
| kernel size | 4,6,8 |
| stride number | 1,2 |
| head number | 2,3,4 |
| head size | 64,128,256 |
| dilation rate | 1,2,4,16 |

ters and 'relu' activation. All filters have kernels of size 4, and move 2 unit-step each time (stride of 2). After this convolutional process, the results are flattened and go through a fully connected layer which maps them into the latent code with pre-defined dimensions. Here we use 8-dimensional latent code for preliminary datasets (original: 24-step) and 16-dimensional vectors for local-global datasets (original: 128-step).

In the decoder, the de-convolutional part has 2 transposed convolutional layers with 128 and 64 filters. All filters have kernels of size 4 and strides of 2 units. Then, the outputs are reshaped and mapped into the original data dimensions (length $L$ and channels $C$) via reshape and fully connected layers, which results in prototypes of the time series. The following Time-Transformer has two blocks, which represents a TCN that has two hidden dilated convolutional layers. The hidden layers have $C$ filters with kernel size of 4 and dilation rate of 1 and 4 respectively. The Transformer blocks combined with each dilated convolutional layer consist of a 3-head self-attention layer with head size of 64 and a feed-forward convolutional layer. The cross attention is also a 3-head attention module with a head size of 64. It takes inputs from different sides which makes it different from the self-attention module ahead. The discriminator, that accomplishes the adversarial process, is a 2-layer fully connected layer with 32 hidden units in both layers and 'relu' activation.

We follow the original AAE training procedure to train the model: We first update the encoder and decoder using the reconstruction loss (Mean Squared Error). Then, the encoder and discriminator are updated with respect to the adversarial loss consisting of a discriminative loss and a generative loss (both are cross-entropy). We use the Adam optimiser to update all the losses. The learning rate for reconstruction loss is 0.005 initially, and reduces to 0.0001 via a polynomial decay function (we directly implemented it using the tensorflow platform, the power of the polynomial is 0.5). Both the discriminative loss and generative loss have initial learning rates of 0.001, which also reduces to 0.0001 via a polynomial decay function (same as previous). The default training epoch is set to 1000.

## D    GENERATION EXAMPLES

We show some generated samples (together with the original data) of local-global datasets in Figure 5.

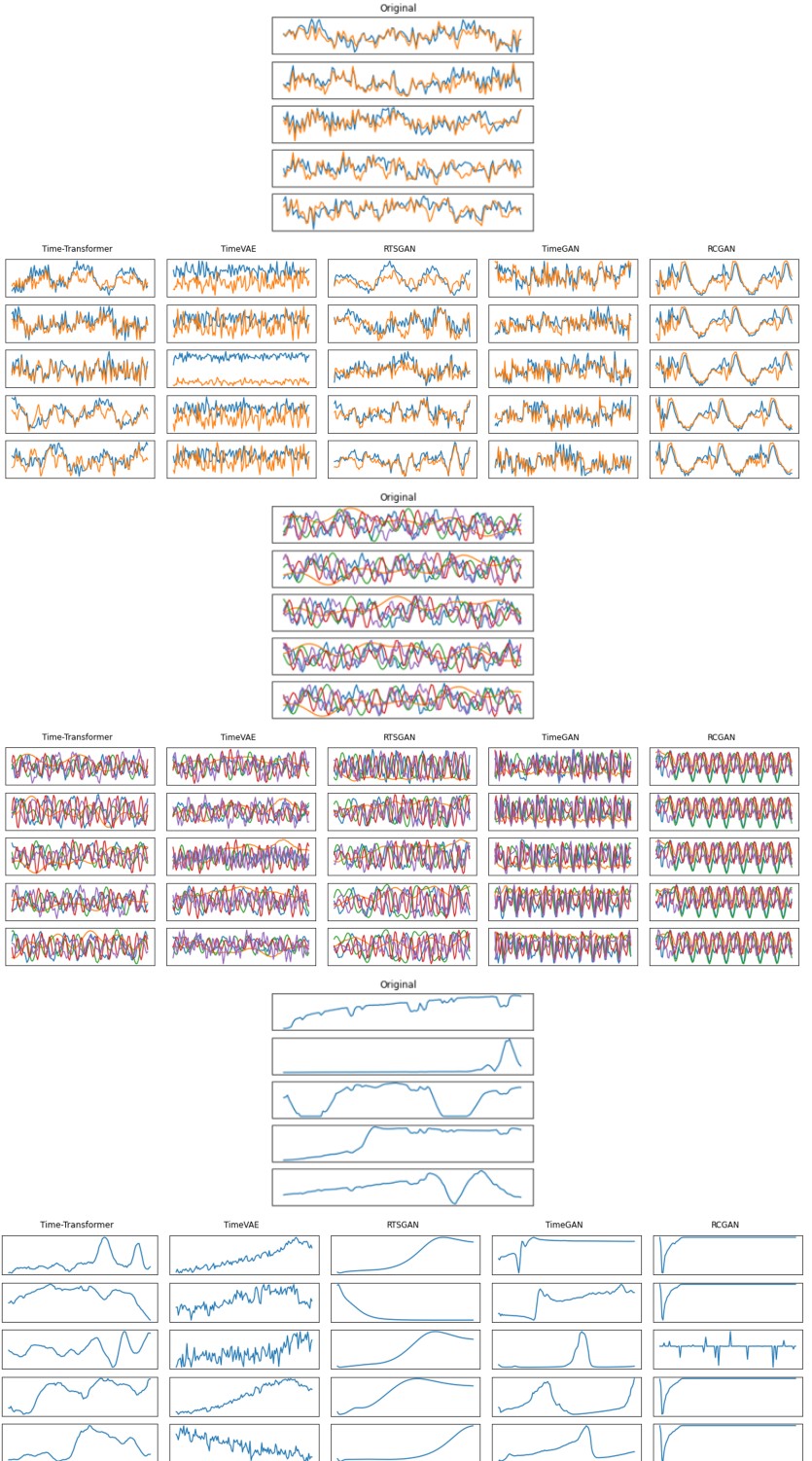

Figure 5: Generation examples

# E ADDITIONAL EXPERIMENTAL RESULTS

We provide more results from the experiments on visualization, training size, ablation study and model application as mentioned in the paper.

## E.1 VISUALIZATION

t-SNE plots of the preliminary datasets are shown in Figure 6.

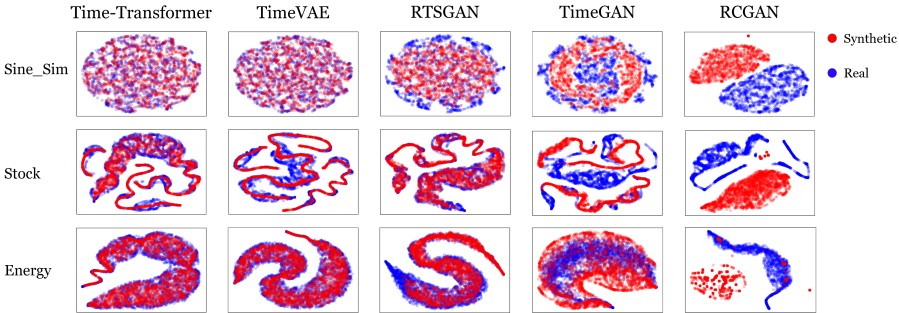

Figure 6: t-SNE visualizations on preliminary datasets

## E.2 ADDITIONAL ABLATION STUDY

Table 7 shows the results of the ablation study with respect to 'Sine_Cpx' and 'ECochG' datasets. Both show performance improvements when we add each part.

Table 7: Extra Ablation Study

| Dataset | Components | FID | Discriminator | Predictor |
|---|---|---|---|---|
| Sine_Cpx | De-Conv | 6.976±0.157 | 0.490±0.021 | 1.832±0.526 |
| | TCN | 2.570±0.067 | 0.417±0.270 | 0.529±0.057 |
| | Transformer | 2.472±0.079 | 0.397±0.253 | 0.484±0.025 |
| | TCN+Trans (Sequential) | 1.721±0.055 | 0.193±0.181 | 0.037±0.013 |
| | Time-Transformer | 1.502±0.062 | 0.168±0.041 | 0.032±0.006 |
| ECochG | De-Conv | 0.781±0.099 | 0.461±0.048 | 0.092±0.011 |
| | TCN | 0.502±0.041 | 0.288±0.192 | 0.027±0.013 |
| | Transformer | 0.513±0.045 | 0.256±0.164 | 0.027±0.007 |
| | TCN+Trans (Sequential) | 0.402±0.031 | 0.162±0.059 | 0.017±0.005 |
| | Time-Transformer | 0.348±0.024 | 0.104±0.012 | 0.013±0.006 |

## E.3 IMBALANCED CLASSIFICATION

Table 8 lists imbalanced classification results on UCR datasets: Wafer, Herring, and SwedishLeaf.

## E.4 TRAINING SIZE COMPARISON

We study how Time-Transformer performs with different sizes of training data. We train the model with 100%, 50%, and 20% of the data. The results are shown in Figure 7, including the mean scores (red dots) and corresponding intervals. Additionally, the best results from competitors are shown via a blue area, with the mean scores represented by the lines in the middle. Note that these results are from the models trained on full size datasets (100%). Generally, a smaller training size leads to worse performance both in terms of quality and stability. The fact that our model is competitive with other models even when trained on 50% of the data highlights its accuracy and stability.

Table 8: Data Augmentation Evaluation on UCR datasets

| Datasets | Components | Accuracy | Recall | Precision | AUC_ROC | AUC_PRC |
|---|---|---|---|---|---|---|
| Wafer | No Aug | 0.9919 | 0.9639 | 0.9610 | 0.9981 | 0.9826 |
| | Replication | 0.9924 | 0.9835 | 0.9478 | 0.9979 | 0.9855 |
| | Jittering | 0.9933 | 0.9774 | 0.9643 | 0.9985 | 0.9892 |
| | RTSGAN | 0.9940 | 0.9759 | **0.9686** | 0.9973 | 0.9847 |
| | TimeVAE | 0.9932 | 0.9789 | 0.9587 | 0.9978 | 0.9868 |
| | **Time-Transformer** | **0.9945** | **0.9849** | 0.9618 | **0.9992** | **0.9928** |
| Herring | No Aug | 0.5469 | 0.3461 | 0.5625 | 0.6549 | 0.5762 |
| | Replication | 0.6563 | **0.4615** | 0.6000 | 0.7287 | 0.6429 |
| | Jittering | 0.6250 | 0.3846 | 0.5556 | 0.6781 | 0.5455 |
| | RTSGAN | 0.6563 | 0.4231 | 0.6111 | 0.7146 | 0.6307 |
| | TimeVAE | 0.6406 | 0.3846 | 0.5882 | 0.6817 | 0.5891 |
| | **Time-Transformer** | **0.6875** | **0.4615** | **0.6667** | **0.7712** | **0.6455** |
| SwedishLeaf | No Aug | 0.9264 | 0.0000 | 0.0000 | 0.9383 | 0.4296 |
| | Replication | 0.8848 | 0.8043 | 0.3700 | 0.9487 | 0.5337 |
| | Jittering | 0.9279 | 0.3043 | 0.5185 | 0.9426 | 0.5230 |
| | RTSGAN | 0.9232 | 0.1956 | 0.4500 | 0.9418 | 0.4656 |
| | TimeVAE | 0.9360 | 0.6304 | 0.5577 | 0.9439 | 0.5433 |
| | **Time-Transformer** | **0.9408** | **0.8478** | **0.5652** | **0.9536** | **0.6341** |

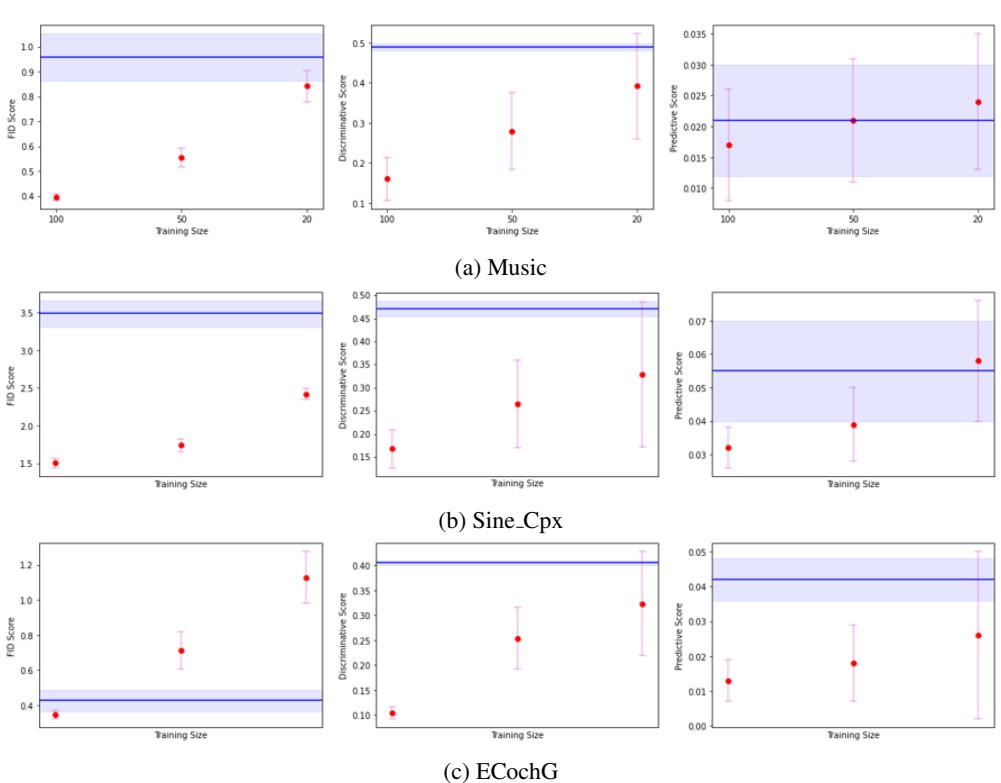

(a) Music

(b) Sine_Cpx

(c) ECochG

Figure 7: Results of training size experiments on 'Music', 'Sine_Cpx', and 'ECochG' datasets

## E.5 CASE STUDY FOR LOCAL & GLOBAL FEATURE LEARNING

We first define the local and global features: Assume we have a time series $x = \{t_1, \ldots, t_n\}$ and a function $F$ that maps a segment of $x$ into some feature space. Global features are defined by applying the function $F$ to the whole time series: i.e. $F(\{t_1, \ldots, t_n\})$. Local features are defined with reference to a sub-part of the whole series: i.e $F(\{t_i, \ldots, t_{i+m}\})$ where, $m < n$.

Then, we use three simple datasets to show different performance of the models when encountering time series containing both global and local features. Figure 8 shows the underlying idea. The green series and the orange series each represent local and global features respectively. They are generated with high (50Hz) and low (5Hz) frequencies with respect to the Fast Fourier Transform. By adding them together we get the blue series which, hence, contains both global and local features, we name it 'mixture' (the global features would be the total trend of the low frequency wave and this needs to be learnt using the entire series, and local features would be the time span and the amplitude of a single oscillation of the high frequency wave, which would be learnt via sub-parts of the time series). To be specific, the formulae to generate local, global and mixture time series are:

- Local: $\alpha sin(2\pi 50t)$
- Global: $\alpha sin(2\pi 5t)$
- Mixture: $\alpha sin(2\pi 5t) + \alpha sin(2\pi 50t)$

where $\alpha \sim \mathcal{U}[1,3]$. For each of the three cases, we generate 2500 time series, each consisting of 128-step time steps (we use 128 time steps to ensure global features are non-trivial). We thus obtain three different datasets, each containing 2500 time series.

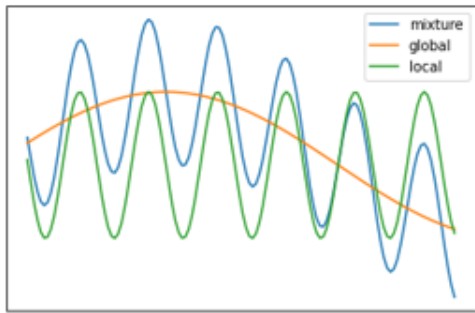

Figure 8: Global & local feature example

We train the different generation models on these datasets and compare their performance. First, we show some generated examples in Figure 9. Comparing these generated examples, we can see: when the time series contain only global features or only local features, all the models can generate comparatively good time series (see Figure 9a and 9b). However, when it comes to the mixture of the two, baseline models fail to generate as effectively. RTSGAN can capture the global trends but fails to extract local properties. TimeGAN and RCGAN seem to be able to learn the local patterns to some extent, but they fail to capture the global features. TimeVAE performs better than others but is less effective than our model (see Figure 9c).

Table 9 lists the quantitative evaluations results of each model, which also shows the effectiveness of our model in extracting both global and local features simultaneously, as it gets the best scores with respect to all metrics for the mixture dataset.

### E.6 CHOICE OF ENCODER

Table 10 lists different performance outcomes from our model with two different encoders, namely, CNN and Time-Transformer. The first is the simple design which we used in our proposed model, and the second is the one with a Time-Transformer module inserted in the encoder. We test their performance on the Sine_Sim dataset to briefly investigate how much improvement a complex encoder can bring to the model. As shown in the table, using a Time-Transformer encoder does improve the results a little, but it requires much more time to train. This indicates that a simple encoder can generate relatively good synthetic data with high efficiency.

### E.7 LONGER TIME SERIES DATA

Most existing related works Yoon et al. (2019); Desai et al. (2021); Pei et al. (2021); Jarrett et al. (2021) for synthetic time series generation have used 24-length time series as their experimental

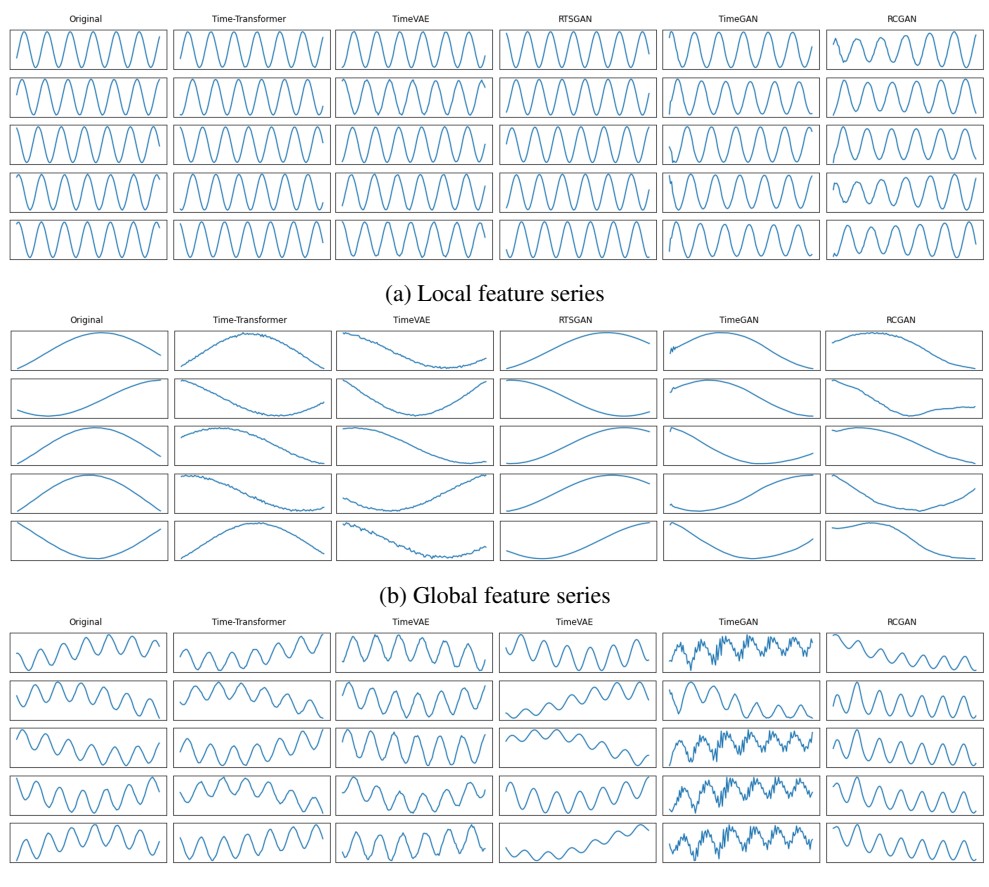

(a) Local feature series

(b) Global feature series

(c) Mixture series

Figure 9: Generation examples

Table 9: Evaluation results for global & local features learning (lower scores are better)

| Model | Benchmark | Local | Global | Mixture |
|---|---|---|---|---|
| RCGAN | | 8.879±0.467 | 4.817±0.592 | 10.217±0.762 |
| TimeGAN | | 4.736±0.224 | 1.597±0.294 | 5.195±0.316 |
| TimeVAE | FID | **0.271±0.074** | 0.241±0.037 | 1.081±0.068 |
| RTSGAN | | 0.299±0.068 | **0.163±0.055** | 1.788±0.091 |
| **Time-Transformer** | | 0.317±0.071 | 0.229±0.019 | **0.474±0.014** |
| RCGAN | | 0.455±0.026 | 0.395±0.011 | 0.497±0.005 |
| TimeGAN | | 0.178±0.062 | 0.048±0.007 | 0.485±0.011 |
| TimeVAE | Discriminative | **0.015±0.026** | 0.018±0.009 | 0.177±0.073 |
| RTSGAN | | 0.022±0.019 | **0.012±0.004** | 0.294±0.078 |
| **Time-Transformer** | | 0.029±0.027 | 0.017±0.015 | **0.061±0.060** |
| Oracle | | 0.009±0.002 | 0.014±0.008 | 0.024±0.003 |
| RCGAN | | 0.196±0.052 | 0.285±0.057 | 0.216±0.079 |
| TimeGAN | | 0.077±0.018 | 0.052±0.013 | 0.103±0.043 |
| TimeVAE | Predictive | **0.013±0.006** | 0.031±0.005 | 0.052±0.011 |
| RTSGAN | | **0.013±0.006** | **0.027±0.012** | 0.073±0.015 |
| **Time-Transformer** | | 0.014±0.005 | 0.030±0.009 | **0.031±0.007** |

datasets. A recent work (PSA-GAN Jeha et al. (2022)) has claimed to generate long time series and the longest time series used in their work are 256-length. They apply a sampling method from LSTnet Lai et al. (2017) to the electricity datasets, to produce these 256-length time series. Hence,

Table 10: Comparison between different encoders (lower results are better)

| Encoder | Training Time (s) | FID | Discriminative | Predictive |
|---|---|---|---|---|
| CNN | **530.24** | 0.283±0.023 | 0.131±0.021 | 0.051±0.015 |
| Time-Transformer | 1651.33 | **0.256±0.014** | **0.120±0.011** | **0.036±0.010** |

we use this dataset, with the same sampling method as Lai et al. (2017), to further evaluate our model. The results are shown in Table 11 below. We can see our model still performs well with this longer (256-length) time series data, as it has the best performance against the baselines (we only use RTSGAN and TimeVAE).

Table 11: Long time series evaluation (lower is better for all metrics)

| Model | FID | Discriminative | Predictive |
|---|---|---|---|
| Oracle | N/A | | 0.010±0.004 |
| TimeVAE | 1.655±0.621 | 0.415±0.037 | 0.034±0.005 |
| RTSGAN | 0.896±0.147 | 0.397±0.007 | 0.021±0.007 |
| **Time-Transformer** | **0.338±0.058** | **0.057±0.027** | **0.014±0.006** |

### E.8 EVALUATION WITH TRANSFORMER

We replace the LSTM with Transformer and perform the quantitative experiments again, in order to investigate if a stronger sequential model can obtain different results. We re-do the experiments on the local-global datasets and use only the two most recent models (RTSGAN, TimeVAE) as our baselines. The results are shown in Table 12. We can see the transformer is more effective at iden-

Table 12: Two scores with Transformer based model (lower numbers are better)

| Model | Benchmark | Music | Sine_Cpx | ECochG |
|---|---|---|---|---|
| TimeVAE | | 0.500±0.000 | 0.487±0.015 | 0.495±0.003 |
| RTSGAN | Discriminative | 0.498±0.001 | 0.499±0.001 | 0.472±0.014 |
| **TimesFormer** | | **0.384±0.026** | **0.425±0.034** | **0.366±0.041** |
| Oracle | | 0.030±0.001 | 0.040±0.001 | 0.007±0.004 |
| TimeVAE | | 0.135±0.019 | 0.051±0.002 | 0.033±0.008 |
| RTSGAN | Predictive | 0.046±0.002 | 0.079±0.002 | 0.016±0.006 |
| **TimesFormer** | | **0.041±0.002** | **0.042±0.001** | **0.009±0.002** |

tifying differences between real data and synthetic ones, as the discriminative scores become worse than those from the LSTM model. However, our model still out-performs the baselines. Regarding the predictive score, our model still has the most effective forecasting performance (closest to using the original data).

## F DETAILS OF IMBALANCED DATASETS

The original ECochG dataset contains data from 77 patients. When sampled and pre-processed as previously mentioned, this results in 13982 time series including two classes. However, this dataset is extremely imbalanced: only 874 instances are positive, which makes it hard to train a machine learning model. One possible solution is augmenting the minority class to make it balanced. As to the UCR datasets, the first two are imbalanced binary datasets while the last one contains 15 classes, we create an imbalanced situation by assigning class 1 as the positive class and all remaining classes as negative (labeled as 0). Here, we provide statistical details of the datasets used in **Model Application** of Section 4.4. Table 13 shows this information including: number of training data (#Train), number of testing data (#Test), length of each time series (Len), positive rate of training set (Pos_Rate, equals to #Positive/#Train), and number of training data for generative model training (#Generative Training).

Table 13: Details of Imbalanced Datasets

| Dataset | # Train | # Test | Len | Pos_Rate | # Generative Training |
|---|---|---|---|---|---|
| ECochG | 9787 | 4195 | 128 | 0.063 | 874 |
| Wafer | 1000 | 6164 | 152 | 0.097 | 97 |
| Herring | 64 | 64 | 512 | 0.391 | 25 |
| SwedishLeaf | 500 | 625 | 128 | 0.058 | 29 |

