# OpenReview forum: "Time-Transformer AAE: Connecting Temporal Convolutional Networks and Transformer for Time Series Generation"
_ICLR.cc/2023/Conference — Submitted to ICLR 2023_

### Official Review · Reviewer_xKc6 · 2022-10-23

**Confidence:** 3
**Correctness:** 3
**Technical Novelty And Significance:** 2
**Empirical Novelty And Significance:** 2
**Recommendation:** 3

**Clarity, Quality, Novelty And Reproducibility:**

The proposed method is easy-to-follow. The Time-Transformer architecture is inspired by the recent Mobile-Former work (Chen et al., 2021) as mentioned in the paper. It can be considered a temporal extension of the Mobile-Former concept.

**Strength And Weaknesses:**

Strengths:

1- The paper leverages the adversarial autoencoder in the temporal domain.

2- The proposed Time-Transformer block controls access to local and global information via a convolution and a self-attention layer, respectively. This design addresses the main concern of the paper: learning local and global features explicitly. It also seems to perform well in the evaluations.

---

Weaknesses:

1- Generative modeling of time-series data is a vast literature involving a number of areas, such as deterministic autoregressive approaches [1*, 2*], state-space models [3*, 4*], and contrastive learning [7*]. Furthermore, [5*, 6*] augment TCNs with a hierarchy of latent variables to capture short-and long-term dependencies. The discussion in the related work section and the baselines in the experiments ignore this line of research. I can suggest another set of ablations. To show the contribution of the proposed Time-Transformer architecture, other model classes could be used in the decoder (i.e., after the De-Convolutional block).

2- The evaluations are performed on "toy" datasets. Sequences of 24 or 128 steps are rather short and hence easily explainable by a fixed-dimensional latent variable. I think this is a limitation as the entire sequence is encoded via a single, global, fixed-dimensional latent space. It is hard to tell how the model would perform when the task requires modeling longer sequences like sequential image generation.

3- Finally, the performance of the baselines, TimeGAN and RCGAN, looks different from the reported performance in the TimeGAN paper. In the t-SNE plots on the preliminary "Sine_Sim" and the "Stock" datasets (see Appendix E), the baselines perform poorly. However, in the TimeGAN paper, (see Fig. 3 a and b in https://proceedings.neurips.cc/paper/2019/file/c9efe5f26cd17ba6216bbe2a7d26d490-Paper.pdf), the generated and the real data are not distinguishable. Could it be possible that the baseline models have not converged? Could the authors explain the source of the difference?

---

Additional References

[1*] Van den Oord, Aaron, et al. "Conditional image generation with pixelcnn decoders." Advances in neural information processing systems 29 (2016).

[2*] Van Den Oord, Aäron, Nal Kalchbrenner, and Koray Kavukcuoglu. "Pixel recurrent neural networks." International conference on machine learning. PMLR, 2016.

[3*] Chung, Junyoung, et al. "A recurrent latent variable model for sequential data." Advances in neural information processing systems 28 (2015).

[4*] Fraccaro, Marco, et al. "Sequential neural models with stochastic layers." Advances in neural information processing systems 29 (2016).

[5*] Lai, Guokun, et al. "Stochastic wavenet: A generative latent variable model for sequential data." arXiv preprint arXiv:1806.06116 (2018).

[6*] Aksan, Emre, and Otmar Hilliges. "STCN: Stochastic temporal convolutional networks." arXiv preprint arXiv:1902.06568 (2019).

[7*] Jarrett, Daniel, Ioana Bica, and Mihaela van der Schaar. "Time-series generation by contrastive imitation." Advances in Neural Information Processing Systems 34 (2021): 28968-28982.


**Summary Of The Paper:**

This paper introduces a decoder architecture for time-series generation. The proposed model, namely the Time-Transformer, uses both a temporal convolutional network (TCN) and a Transformer to process information at different timescales. While the TCN focuses on learning local features, the Transformer block aims to capture global features, which are then fused via a cross-attention operation. The decoder network stacks several of these Time-Transformer layers. The proposed decoder is trained via the adversarial autoencoder framework to learn a latent space. The model is evaluated on a variety of time series benchmarks (Sine, Stock, and Energy datasets). It demonstrates better performance compared to the baselines.

**Summary Of The Review:**

It is an interesting line of work, and the paper presents strong performance in the evaluations. However, considering the originality of the architecture, I think the paper requires a revision to provide more insights. The related work discussion also misses relevant time-series models. I believe providing more comprehensive evidence for the contribution of the proposed model would be very helpful for the community. The following questions can be a starting point. Is the Time-Transformer also effective in other frameworks (i.e., seq2seq VAE or autoregressive setting instead of the adversarial autoencoders)? How would other decoder classes perform in the benchmarks?

I am leaning towards a rejection for the current version of the submission as it would require a major revision. However, I am willing to change my score upon clarification and new evidence.

---

> ### Author Response · Authors · 2022-11-15
> **Response for Reviewer xKc6 (Part1)**
>
> We would like to thank the reviewer for taking time and effort to give these useful comments, next we will try to address reviewer's concerns point by point:
>
> - Generative modeling of time-series data has a sizeable literature involving a number of areas, such as deterministic auto-regressive approaches [1*, 2*], state-space models [3*, 4*], and contrastive learning [7*]. Furthermore, [5*, 6*] augment TCNs with a hierarchy of latent variables to capture short-and long-term dependencies. The discussion in the related work section and the baselines in the experiments ignore this line of research. I can suggest another set of ablations. To show the contribution of the proposed Time-Transformer architecture, other model classes could be used in the decoder (i.e., after the De-Convolutional block).
>
> Although the works mentioned by the reviewer are also related to the time series generation, they are somewhat different to our task. Our work is most related to data generation via GANs, where the time series samples (it could be an entire time series or a sampled segment via sampling methods such as a sliding window) are used during training for the model to learn the distribution of the entire dataset. Via adversarial training, the model will learn to match the prior distribution to the distribution of data (or latent variables of the original data), so that during sampling step, it can generate synthetic data purely from the prior distribution (without any information about the time series data). In this line, the generated data needed to be: 1. distributed close to the original data, 2. preserve the temporal property of the original data. Hence we can use evaluation metrics assessing these properties.
>
> However, [1*-6*] focus on another type of generative model, named auto-regressive model, where data is generated step-by-step. Specifically, references [1*,2*] are interesting image data generation works, which use auto-regressive methods to generate image pixels in order to accomplish an image inpainting task. However they did not include any study in the context of time series. Extending them to time series could be interesting future work. References [3*-6*] discussed sequential data generation with the auto-regression basis. They utilize the variational inference of the VAE at each time step, so that the model can generate the future steps based on the previous steps and the ground truth data. Unlike models in our line, these models require part of the original data to be available for the sampling step.
>
> We think this is the main reason why existing works (such as [7*,R1,R2,R3,R4]) have not mentioned or compared against [3*-6*]. Nonetheless, we acknowledge this line of work is interesting in the context of time series data generation and may have potential to further improve the time series generation works when combined with the existing models like ours. Thus, we have expanded the paper's literature review to include these works. Please see section 2 of the revised paper.
>
> We note that [7*] is in the same line as our work, which is generating fake time series from the learnt distribution with a view to better data augmentation for scenarios where there is a lack of data. Instead of using deep generative models, [7*] proposed a contrastive learning based model. This is an interesting approach for synthetic data generation. We also added it into our literature review (see section 2 of the revised paper). However, we find that the author didn't provide publicly available code. Hence it is impractical to add it as an additional baseline.
>
> Ref:
>
> [1*] Van den Oord, Aaron, et al. "Conditional image generation with pixelcnn decoders." Advances in neural information processing systems 29 (2016).
>
> [2*] Van Den Oord, Aäron, Nal Kalchbrenner, and Koray Kavukcuoglu. "Pixel recurrent neural networks." (2016)
>
> [3*] Chung, Junyoung, et al. "A recurrent latent variable model for sequential data." (2015).
>
> [4*] Fraccaro, Marco, et al. "Sequential neural models with stochastic layers." (2016).
>
> [5*] Lai, Guokun, et al. "Stochastic wavenet: A generative latent variable model for sequential data." (2018).
>
> [6*] Aksan, Emre, and Otmar Hilliges. "STCN: Stochastic temporal convolutional networks." (2019).
>
> [7*] Jarrett, Daniel, Ioana Bica, and Mihaela van der Schaar. "Time-series generation by contrastive imitation." Advances in Neural Information Processing Systems 34 (2021): 28968-28982.
>
> [R1] Paul Jeha, et al. "PSA-GAN: Progressive self attention GANs for synthetic time series." (2022)
>
> [R2] Abhyuday Desai, et al. "Timevae: A variational autoencoder for multivariate time series generation." (2021)
>
> [R3] Jinsung Yoon, et al. "Time-series Generative Adversarial Networks." (2019)
>
> [R4] Hengzhi Pei, et al. "Towards Generating Real-World Time Series Data." (2021)

---

> > ### Comment · Reviewer_xKc6 · 2022-11-23
> > **Thanks for the inputs**
> >
> > I thank the authors for their rebuttal. Although I think that the benchmark datasets are small, I acknowledge that the authors merely followed established benchmarks from peer-reviewed works.
> >
> > However, my concern regarding the performance of the baseline methods still remains. I would like to ask why every paper reports different numbers on the same datasets, such as the energy and stock datasets. The results reported in the TimeVAE and the TimeGAN papers and in this paper are highly different. Isn't it possible to directly take the performances from the corresponding papers and evaluate the proposed method in the same setting? The poor performance of the baseline methods is concerning.

---

> > > ### Author Response · Authors · 2022-11-26
> > > **Thanks for the response**
> > >
> > > We would like to thank the reviewer for the response.
> > >
> > > Regarding the baseline results. We ran all the models under the same environment for a fair comparison. As previously explained, taking into consideration differences potentially caused by different random seeds and hardware platforms, it is unsurprising that different works have obtained different results. Thus, we feel it is better to run all the models again under the same environment rather than directly use the existing results from previous paper.
> > >
> > > Nonetheless, we try our best to address reviewer's concerns. We find the biggest difference lies in the reported predictor scores: ours are much smaller than other related works in general. Thus, we double-checked the code of TimeGAN and find that we had calculated the predictor score in a different manner: we used the common one-step-ahead forecast: assume we have a time series $x=\[t_1, \cdots, t_n \]$, where $t_i = \[ t_i^{(1)}, \cdots, t_i^{(C)} \] ^T$. The inputs are the first $n-1$ time steps of all dimensions ($x_{train} = \[t_1, \cdots, t_{n-1} \] $), and the model learns to predict the last time step ($y_{train} = \[ t_n \]$). The loss is the mean absolute error between the predictions and the ground truth ($| y_{train} - y_{pred} |$).
> > >
> > > However, in TimeGAN, the inputs to the predictor are the first $n-1$ steps of the first $C-1$ dimensions ($x_{train} = \[t_1', \cdots, t_{n-1}' \]$, where $t_i' = \[ t_i^{(1)}, \cdots, t_i^{(C-1)} \]^T$). The model learns to predict the last dimension with one more step ($y_{train} = \[ t_2^{(C)}, \cdots, t_n^{(C)} \]$). The loss is a mean absolute error with respect to the new outputs. What's more, TimeGAN used a sigmoid function as the last activation (while we don't use any activation in the last layer).
> > >
> > > These methodological differences led to the different results of the predictor score between our work and that of TimeGAN. Now, we have also used their method to get a new set of predictor scores. The results are shown in Table below (lower numbers are better).
> > > |     Model          |     Sine_sim       |     Stock          |     Energy         |     Music          |     Sine_Cpx       |     ECochG         |
> > > |--------------------|--------------------|--------------------|--------------------|--------------------|--------------------|--------------------|
> > > |     RCGAN          |     0.277±0.003    |     0.043±0.002    |     0.301±0.004    |     0.143±0.002    |     0.305±0.003    |     0.051±0.008    |
> > > |     TimeGAN        |     0.257±0.004    |     0.040±0.001    |     0.278±0.005    |     0.106±0.003    |     0.234±0.005    |     0.013±0.002    |
> > > |     TimeVAE        |     0.218±0.001    |     0.040±0.000    |     0.254±0.001    |     0.102±0.002    |     0.189±0.001    |     0.011±0.003    |
> > > |     RTSGAN         |     0.242±0.002    |     0.038±0.000    |     0.253±0.005    |     0.097±0.001    |     0.227±0.006    |     0.014±0.002    |
> > > |     TimesFormer    |     0.223±0.002    |     0.039±0.001    |     0.261±0.001    |     0.086±0.001    |     0.187±0.001    |     0.009±0.000    |
> > > |     Original       |     0.215±0.001    |     0.037±0.000    |     0.249±0.002    |     0.080±0.001    |     0.168±0.003    |     0.008±0.000    |
> > >
> > > These predictor scores are similar to the previous works on the corresponding datasets, with acceptable variances taking into account the different random seeds and hardware environments. This provides additional evidence that our baseline models have been implemented properly. On the other hand, we can see although the values have changed, our claim in paper still stands: our model performs the best for the three local-global datasets (Music, Sine\_Cpx, ECochG), which indicates its effectiveness in extracting global and local features simultaneously.
> > >
> > > Given this additional evidence, we think that while some differences in reported results exist between our work and the TimeGAN, likely due to hardware and random seed choices, they don't affect overall conclusions of our paper.

---

> > > > ### Author Response · Authors · 2022-12-05
> > > > **Thanks for the feedback**
> > > >
> > > > Dear reviewer xKc6,
> > > > We would like to thank you again for the time you dedicated to reviewing our paper. We have tried to address your concern on baselines via more explanations and experiments. Since the end of discussion period is getting close, we would appreciate it if you kindly let us know if you still have concerns related to the baselines. We would be willing to try or best to address any further questions.

---

> ### Author Response · Authors · 2022-11-15
> **Response for Reviewer xKc6 (Part2)**
>
> - The evaluations are performed on "toy" datasets. Sequences of 24 or 128 steps are rather short and hence easily explainable by a fixed-dimensional latent variable. I think this is a limitation as the entire sequence is encoded via a single, global, fixed-dimensional latent space. It is hard to tell how the model would perform when the task requires modeling longer sequences like sequential image generation.
>
> Most existing related works [7*,R2,R3,R4] for synthetic time series generation have used 24-length time series as their experimental datasets. In recent work [R1] has claimed to generate long time series and the longest time series used in their work are 256-length. They apply a sampling method from LSTnet [R5] to the electricity dataset (from UCI time series database), to produce these 256-length time series. Hence, we use this dataset, with the same sampling method as [R5], to further evaluate our model.
>
> The results are shown in Table below. Here we use 'Disc.' and 'Pred.' to represent the discriminative score and the predictive score in paper respectively (a quick reminder, the discriminative score is using a classifier to distinguish the real and synthetic data, and the predictive score is training a predictor on synthetic data and test on real data). We can see our model still performs well with this longer (256-length) time series data, as it has the best performance against the baselines. (This has been included as Appendix E7 of the revised paper).
>
> |           Model         |         FID        |        Disc.       |        Pred.       |
> |:-----------------------:|:------------------:|:------------------:|:------------------:|
> |          Oracle         |         N/A        |         N/A        |     0.010±0.004    |
> |          TimeVAE        |     1.655±0.621    |     0.415±0.037    |     0.034±0.005    |
> |          RTSGAN         |     0.896±0.147    |     0.397±0.007    |     0.021±0.007    |
> |     Time-Transformer    |     0.338±0.058    |     0.057±0.027    |     0.014±0.006    |
>
> - Finally, the performance of the baselines, TimeGAN and RCGAN, looks different from the reported performance in the TimeGAN paper. In the t-SNE plots on the preliminary "Sine\_Sim" and the "Stock" datasets (see Appendix E), the baselines perform poorly. However, in the TimeGAN paper, the generated and the real data are not distinguishable. Could it be possible that the baseline models have not converged? Could the authors explain the source of the difference?
>
> Since the official codes of TimeGAN do not specify the random seeds, there is a high possibility that the results may be different from the original ones reported in the TimeGAN paper. Hardware platform differences may also contribute to differences in reported results [8, 12, 13]. We note that in the TimeVAE paper, they also reported different results for the baselines TimeGAN and RCGAN on the mentioned datasets (see Figure 4 in [R2]). We thus believe such differences may be unsurprising due to use of differing random seeds and differing hardware platforms.
>
> Ref:
>
> [R5] Guokun Lai, et al. "Modeling Long- and Short-Term Temporal Patterns with Deep Neural Networks." (2017)
>
> [R6] Geir Kjetil Sandve, et al. "Ten Simple Rules for Reproducible Computational Research." (2013)
>
> [R7] https://machinelearningmastery.com/different-results-each-time-in-machine-learning/
>
> [R8] https://machinelearningmastery.com/reproducible-results-neural-networks-keras/

---

### Official Review · Reviewer_GBz2 · 2022-10-24

**Confidence:** 3
**Correctness:** 3
**Technical Novelty And Significance:** 2
**Empirical Novelty And Significance:** 2
**Recommendation:** 5

**Clarity, Quality, Novelty And Reproducibility:**

This paper is written in good quality & clarity, main methods and designs are well presented in the Introduction, Method and Experiments section. Figures & tables are very useful for understanding the overall design. Experiments covered most aspects of time series generation, and ablation study clearly demonstrated the impact of the cross-attention layer. On originality, the use of TCN and Transformer are common in time series applications, the originality comes only from the design bidirectional cross attention. Therefore, the originality is limited.


**Strength And Weaknesses:**

Generally this paper does well in presenting its idea and performance, but struggles to provide a coherent story to help us understand how the idea was developed and how the claim of better capturing local correlation & global pattern is supported.

Strengths of this paper:
1. The figures are very informative in demonstrating the model architecture & bidirectional cross attention.
2. Extensive experiments are conducted over multiple aspects to understand the performance difference between the proposed model and multiple baselines.
3. The experiment results look very promising, huge improvement over previous state-of-the-art.
4. Ablation study demonstrated the proposed bidirectional cross attention mechanism is very useful.

Weaknesses and how to improve:
1. The datasets are a bit limited and on a small scale. Can you try a larger dataset like [eletricity](https://archive.ics.uci.edu/ml/datasets/ElectricityLoadDiagrams20112014) or [traffic](https://archive.ics.uci.edu/ml/datasets/PEMS-SF) and see if there's any difference in the results?
2. While the paper claimed to better capture local correlation and global dependency, there's no clear analysis to support this claim. Case study or explanation with characteristics of the dataset and performance could mitigate this gap.
3. The discriminative score and predictive score are generated with a weak baseline (2 layer LSTM), in real applications people use more sophisticated methods like a Transformer based model to do classification & prediction. Would you please try better classifiers or predictors and see if the result changes? This is because these new methods do better in capturing subtle long term dependency and local patterns.
4. The motivation of using the proposed architectures is not well explained. There're more choices in how to capture local / long term dependency. Would you please add some motivations to help me understand the necessities in using the proposed architecture?


**Summary Of The Paper:**

This paper introduced a new time series generative model called `Time-Transformer AAE` that consists of an adversarial autoencoder and a newly designed architecture called `Time-Transformer`, where temporal properties are learnt by both a TCN layer and a Transformer block. This model was proposed to better learn local correlations as well as global dependencies. Empirical results demonstrated that this model can achieve state-of-the-art performance on datasets, especially datasets with long term dependencies.

The contributions of this paper comes from the creation of this Transformer based adversarial auto-encoder model, proposing the layer-wise parallel design & interaction to simultaneously learn local & global features, and demonstrating its performance via empirical experiments.


**Summary Of The Review:**

Overall, this paper proposed an interesting idea of combining TCN & Transformer in a Time-Transformer block for adversarial autoencoder time series generation. The presentation of model architecture and performance are clear. On the other hand, the motivation of such a proposed model is a bit unclear, and the application of such a model in new areas seems limited. It also appears unclear to me whether the performance gain comes from combining different layers of previous state-of-the-art model layers or it's a genuine change. Therefore, I recommend this paper to be rejected.

---

> ### Author Response · Authors · 2022-11-15
> **Response for Reviewer GBz2 (P2)**
>
> 3.We use the transformer to do these experiments again. Considering time limitations, we only re-did the experiments on the local-global datasets and compared to only the two most recent models (RTSGAN, TimeVAE). The results are shown in Table below. (This been included as Appendix E8 of the revised paper). We can see the transformer is more effective at identifying differences between real data and synthetic ones, as the discriminative scores become worse than those from LSTM model. However, our model is still more effective than the baselines. Regarding the predictive score, our model still has the most effective forecasting performance (closest to using the original data).
>
> |           Model         |     Benchmark    |        Music       |       Sine_Cpx     |        ECochG      |
> |:-----------------------:|:----------------:|:------------------:|:------------------:|:------------------:|
> |          TimeVAE        |       Disc.      |     0.500±0.000    |     0.487±0.015    |     0.495±0.003    |
> |          RTSGAN         |         Disc.         |     0.498±0.001    |     0.499±0.001    |     0.472±0.014    |
> |     Time-Transformer    |         Disc.         |     0.384±0.026    |     0.425±0.034    |     0.366±0.041    |
> |          Oracle         |       Pred.      |     0.030±0.001    |     0.040±0.001    |     0.007±0.004    |
> |          TimeVAE        |         Pred.         |     0.135±0.019    |     0.051±0.002    |     0.033±0.008    |
> |          RTSGAN         |       Pred.       |     0.046±0.002    |     0.079±0.002    |     0.016±0.006    |
> |     Time-Transformer    |          Pred.        |     0.041±0.002    |     0.042±0.001    |     0.009±0.002    |
>
> 4.We designed this architecture mainly to address the problem of learning local and global features of time series data simultaneously, which has not been studied previously. Regarding the choice of the model, some previous studies [7,8,9,10] show that CNN based models are good at local processing and Transformer based models are good at learning global features. We used this insight as evidence that it would be effective to combine two models together, to learn local and global features simultaneously. We acknowledge that there exist other famous models that might be suitable for extracting local or global features and these of course could be studied in a future work.
>
> Regarding the architecture of our model, since we want the model to learn two different levels (local and global) of features simultaneously, it is necessary to design a parallel structure that enables the model to learn independent features at the same time. Evidence for the effectiveness of this choice can be found in the Ablation study in section 4.4 of the revised paper, Table 4 of this section shows the source of the improvement. The last two rows compare the simple combination of the two models versus our designed architecture, and we see that our design has better  performance.
>
> Ref:
>
> [7] Rajat Sen, Hsiang-Fu Yu, and Inderjit S Dhillon. "Think globally, act locally: A deep neural network approach to high-dimensional time series forecasting." (2019)
>
> [8] Patrick Esser, et al. "Taming Transformers for High-Resolution Image Synthesis." (2020)
>
> [9] Wentian Zhao, et al. "Deep temporal convolutional networks for short-term traffic flow forecasting." (2019)
>
> [10] Yinpeng Chen, et al. "Mobile-former: Bridging mobilenet and transformer." (2021)

---

> > ### Author Response · Authors · 2022-12-05
> > **Thanks for the feedback**
> >
> > Dear Reviewer GBz2,
> > We would like to thank you again for the time you dedicated to reviewing our paper. We have tried to address your concerns via more explanations and experiments. Since the end of discussion period is getting close and we have not heard back from you yet, we would appreciate it if you kindly let us know if you still have concerns related to the mentioned issues such as the case study to support global-local features learning ability and the motivation of the model design. We would be willing to address any further questions.

---

> ### Author Response · Authors · 2022-11-15
> **Response for Reviewer GBz2 (P1)**
>
> We would like to thank the reviewer for taking time and effort to give these useful comments, next we will try to address reviewer's concerns point by point:
>
> 1.Most existing related works [1,2,3,4] for synthetic time series generation have used 24-length time series as their experimental datasets. A recent work [5] has claimed to generate long time series and the longest time series used in their work are 256-length. They  apply a sampling method from LSTnet [6] to the electricity datasets, to produce these 256-length time series . Hence, we use this datasets, with the same sampling method as [6], to further evaluate our model. The results are shown in Table below. Here we use 'Disc.' and 'Pred.' to represent the discriminative and predictive score respectively. We can see our model still performs well with this longer (256-length) time series data, as it has the best performance against the baselines. (This has been included as Appendix E7 of the revised paper).
> |           Model         |         FID        |        Disc.       |        Pred.       |
> |:-----------------------:|:------------------:|:------------------:|:------------------:|
> |          Oracle         |         N/A        |         N/A         |     0.010±0.004    |
> |          TimeVAE        |     1.655±0.621    |     0.415±0.037    |     0.034±0.005    |
> |          RTSGAN         |     0.896±0.147    |     0.397±0.007    |     0.021±0.007    |
> |     Time-Transformer    |     0.338±0.058    |     0.057±0.027    |     0.014±0.006    |
>
> 2.To better illustrate, we have added an extra case study in Appendix E.5 of the revised paper.  This case study examines a simple scenario where we generate three simple time series including a local (high frequency), a global (low frequency), and a mixture of them. This is done based on Fourier decomposition. To be specific, the formulas to generate local, global and mixture time series are
>
> * Local: $\alpha sin(2\pi 50 t)$
>
> * Global: $\alpha sin(2\pi 5 t)$
>
> * Mixture: $\alpha sin(2\pi 5 t) + \alpha sin(2\pi 50 t)$
>
> where $\alpha \sim \mathcal{U}[1,3]$. For each of the three cases, we generate 2500 time series, each consisting of 128-step time steps (we use 128 time steps to ensure global features are non-trivial).  We thus obtain three different datasets, each containing 2500 time series. Through experiments, we find when the time series contains only global or local feature, all the models can generate comparative good time series. However, when it comes to the mixture of two, baseline models fail to generate good synthesis. The RTSGAN can capture the global trends but fail to extract local properties. TimeGAN and RCGAN seem to be able to learn the local patterns to some extent, but they fail to capture the global features. TimeVAE performs better than others but still less effective than our model.
>
> Ref:
>
> [1] Jarrett, Daniel, Ioana Bica, and Mihaela van der Schaar. "Time-series generation by contrastive imitation." (2021)
>
> [2] Abhyuday Desai, et al. "Timevae: A variational autoencoder for multivariate time series generation." (2021)
>
> [3] Jinsung Yoon, et al. "Time-series Generative Adversarial Networks." (2019)
>
> [4] Hengzhi Pei, et al. "Towards Generating Real-World Time Series Data." (2021)
>
> [5] Paul Jeha, et al. "PSA-GAN: Progressive self attention GANs for synthetic time series." (2022)
>
> [6] Guokun Lai, et al. "Modeling Long- and Short-Term Temporal Patterns with Deep Neural Networks." (2017)

---

### Official Review · Reviewer_Cnjw · 2022-10-25

**Confidence:** 3
**Correctness:** 3
**Technical Novelty And Significance:** 3
**Empirical Novelty And Significance:** 3
**Recommendation:** 6

**Clarity, Quality, Novelty And Reproducibility:**

The writing and presentation is clear. The "weaknesses" section includes some items that I think detract from the quality and limit the overall novelty of the method. However, the quality of the experiments and results appears to be good. Code is provided, but reproducibility would be improved by including details of the training algorithm in the paper itself.

**Strength And Weaknesses:**

Strengths:
- The paper is well written and the main ideas are clearly presented. The figures and supplementary details are relevant and informative.
- The model is studied through a broad set of experiments that show it consistently outperforms recent, relevant competitor methods in terms of some standard quality metrics for generative modes: predictive value of the synthetic data for a real test set, difficulty of discriminating between real and generated sequences, and overlap of the empirical and generative distributions.
- The ablation study (Table 4) provides convincing evidence that the specifically parallel structure of the time-transformer block is key to achieving the results above. This goes some way towards alleviating weakness #1 below.

Weaknesses:
- The novelty of the proposal is modest overall. The specific innovation here is the parallel arrangement of two well-known architectures as part of a decoder block in a well-known generative framework.
- Throughout the paper, the authors repeatedly claim that simultaneous learning of local and global structure in time series is a major challenge that is specifically overcome by their approach. Despite this, this problem is not defined at any level of technical detail, the property is not established to obtain with any level of rigor in the provided data, and there is no evidence provided the superior performance of the proposed method is due to solving this specific problem.
- There are no training details whatsoever in the main paper. The supplement describes the learning rate schedule but not the algorithmic details of training. The algorithmic details of training need to be included somewhere in the paper or supplement.

**Summary Of The Paper:**

The authors introduce Time-Transformer AAE, a generative model for time series that incorporates a new "time-transformer" architecture into the adversarial autoencoder (AAE) generative framework. The time-transformer consists of parallel temporal convolutional and transformer blocks interleaved with bidirectional cross-attention to share information. Empirical studies on a variety of datasets show that this approach achieves good coverage and captures the predictive information in the original data, while ablations show that the specific architecture of the time-transformer block outperforms reasonable alternatives.

**Summary Of The Review:**

The authors propose an architecture and framework for generative time series modeling that borrows significantly from previous work, yet is shown to perform well relative to state-of-the-art competitors in a broad range of experiments. Moreover, the quality of these results seems to depend specifically on the novel way in which these familiar components are combined. Other issues in the presentation would be straightforward to address and could improve my overall evaluation of the work.

---

> ### Author Response · Authors · 2022-11-15
> **Response for Reviewer Cnjw**
>
> We would like to thank the reviewer for taking time and effort to give these useful comments, next we will try to address reviewer's concerns point by point:
>
> - The novelty of the proposal is modest overall. The specific innovation here is the parallel arrangement of two well-known architectures as part of a decoder block in a well-known generative framework.
>
> Our main contributions in this paper lie not only in the parallel design, but also in the connections between the two sub-models: the proposed bidirectional cross attention is needed to effectively link between the different types of features (local and global). Our designed model properly learns local and global features simultaneously, which is a gap for existing methods. From this perspective, we believe our proposed approach is a significant innovation and we have provided a range of experimental evidence for its effectiveness.  Additionally, we also showed in the ablation study that simply linearly combining two models leads to a worse performance than our model (last two rows in Table 4 of the revised paper).
>
> - Throughout the paper, the authors repeatedly claim that simultaneous learning of local and global structure in time series is a major challenge that is specifically overcome by their approach. Despite this, this problem is not defined at any level of technical detail, the property is not established to obtain with any level of rigor in the provided data, and there is no evidence provided the superior performance of the proposed method is due to solving this specific problem.
>
> Assume we have a time series $x=\{t_1, \dots, t_n \}$ and a function $F$ that maps a segement of $x$ into some feature space. Global features are defined by applying the function to $F$ to the whole time series: i.e. $F(\{t_1, \dots, t_n\})$. Local features are defined with reference to a sub-part of the whole series: i.e $F(\{t_i, \dots, t_{i+m} \})$.
>
> To better illustrate, we have added an extra case study in Appendix E.5 of the revised paper.  This case study examines a simple scenario where we generate three simple time series including a local (high frequency), a global (low frequency), and a mixture of them. This is done based on Fourier decomposition (in the mixture, the global features would be the total trend of the low frequency wave and this needs to be learnt using the entire series, and local features would be the time span and the amplitude of a single oscillation of the high frequency wave, which would be learnt via sub-parts of the time series). To be specific, the formulas to generate local, global and mixture time series are:
>
> *Local: $\alpha sin(2\pi 50 t)$
>
> *Global: $\alpha sin(2\pi 5 t)$
>
> *Mixture: $\alpha sin(2\pi 5 t) + \alpha sin(2\pi 50 t)$
>
> where $\alpha \sim \mathcal{U}[1,3]$. For each of the three cases, we generate 2500 time series, each consisting of 128-step time steps (we use 128 time steps to ensure global features are non-trivial).  We thus obtain three different datasets, each containing 2500 time series.
>
> Through experiments, we find when the time series contains only global or local features, all the models can generate comparative good time series. However, when it comes to the mixture of two, baseline models fail to generate good synthesis. The RTSGAN can capture the global trends but fails to extract local properties. TimeGAN and RCGAN seem to be able to learn the local patterns to some extent, but they fail to capture the global features. TimeVAE performs better than others but is still less effective than our model.}
>
> - There are no training details whatsoever in the main paper. The supplement describes the learning rate schedule but not the algorithmic details of training. The algorithmic details of training need to be included somewhere in the paper or supplement.
>
> We added the following details of the training procedure in the last paragraph of Appendix C.2 of the revised paper: ``We follow the original AAE training procedure to train the model: We first update the encoder and decoder using the reconstruction loss (Mean Squared Error). Then, the encoder and discriminator are updated with respect to the adversarial loss consisting of a discriminative loss and a generative loss (both are Cross-entropy). We use the Adam optimiser to update all the losses. The learning rate for reconstruction loss is 0.005 initially, and reduces to 0.0001 via a polynomial decay function (we directly implemented it using the tensorflow platform, the power of the polynomial is 0.5). Both the discriminative loss and generative loss have initial learning rates of 0.001, which also reduces to 0.0001 via a polynomial decay function (same as previous). The default training epoch is set to 1000.''

---

> > ### Author Response · Authors · 2022-12-05
> > **Thanks for the feedback**
> >
> > Dear reviewer Cnjw,
> > Thanks very much for your recognition of our work and the encouraging comments. They have greatly improved our work. Please feel free to let us know if you have more questions.

---

### Official Review · Reviewer_Kd3s · 2022-10-25

**Confidence:** 3
**Correctness:** 3
**Technical Novelty And Significance:** 3
**Empirical Novelty And Significance:** Not applicable
**Recommendation:** 6

**Clarity, Quality, Novelty And Reproducibility:**

Clarity: the writing is clear and easy to follow.

Quality: the proposed approach looks technically sound, and the evaluation is comprehensive. However, some

Novelty: good.

Reproducibility: good since implementation details are provided.

**Strength And Weaknesses:**

Strengths:
1. The proposed Time-Transformer is a novel architecture for time series generation.
2. Experimental results show that the proposed method could significantly improve the results.
3. Writing is clear and easy to follow.


Weaknesses and questions:
1. How will your model perform if you use GAN rather than the adversarial auto-encoder framework?
2. The encoder is much simpler than the decoder, what's the motivation behind your choice?
3. How does Time-Transformer perform if you use two parallel transformer blocks rather than a TCN and a transformer?

**Summary Of The Paper:**

This paper introduces Time-Transformer AAE. The proposed method is an adversarial autoencoder approach, and the Time-Transformer is a component of the decoder. Each Time-Transformer block is comprised of (1) two parallel modules: a TCN block and a transformer block, and (2) a bi-directional cross-attention. The experimental results show that the proposed method could significantly outperform SOTA methods.

**Summary Of The Review:**

In general, this paper introduces a novel method for time series generation, and the results show that it can significantly outperform the SOTA methods. However, the motivations for some choices and designs are not quite clear (please see weaknesses). In my opinion, the strengths outweigh the weaknesses.

---

> ### Author Response · Authors · 2022-11-15
> **Response for Reviewer Kd3s**
>
> We would like to thank the reviewer for taking time and effort to give these useful comments, we address the reviewer's concerns point by point:
>
> 1.How will your model perform if you use GAN rather than the adversarial auto-encoder framework?
>
> Theoretically a GAN backbone will perform similarly to an AAE based model [1]. The original AAE can be regarded as a combination of the VAE and GAN. It replaces the variational inference in VAE with the adversarial training from GAN. This makes AAE able to produce similar outcomes to a GAN, but with higher efficiency, since the adversarial procedure is now implemented in the latent space rather than the original data space.
>
> 2.The encoder is much simpler than the decoder, what's the motivation behind your choice?
>
> Our design adds a Time-Transformer block into the basic AAE to refine the learning outcomes of this basic model. Considering our focus is about the generated data rather than the latent code, we put the new block after the de-convolution and model the encoder using a simple design to save computation. We have now performed a simple case study to compare different types of encoders using Sine\_Sim dataset. As shown in Table  below. Here we use 'Disc.' and 'Pred.' to represent the discriminative score and the predictive score in paper respectively (a quick reminder, the discriminative score is using a classifier to distinguish the real and synthetic data, and the predictive score is training a predictor on synthetic data and test on real data). Using a time-transformer encoder does improve the results a little, but it requires much more time to train. This indicates that a simple encoder can generate relatively good synthetic data with high efficiency. (This has been included as Appendix E6 in the revised paper).
>
> |          Encoder        |     Training Time    |         FID        |        Disc.       |        Pred.       |
> |:-----------------------:|:--------------------:|:------------------:|:------------------:|:------------------:|
> |            CNN          |         530.24       |     0.283±0.023    |     0.131±0.021    |     0.051±0.015    |
> |     Time-Transformer    |        1651.33       |     0.256±0.014    |     0.120±0.011    |     0.036±0.010    |
>
> 3.How does Time-Transformer perform if you use two parallel transformer blocks rather than a TCN and a transformer?
>
> Our design  leverages the advantages of TCN for local feature extraction and Transformer for global feature learning, so that the model will be able to learn both feature types simultaneously. Using two transformers would lead the model to learn the global features twice while ignoring the local properties. This would not follow our objective of learning both local and global properties.
>
> [1] Alireza Makhzani, et al. Adversarial Autoencoders. (2015)

---

> > ### Author Response · Authors · 2022-12-05
> > **Thanks for the feedback**
> >
> > Dear reviewer Kd3s,
> > Thanks very much for your recognition of our work and the encouraging comments. They have greatly improved our work. Please feel free to let us know if you have more questions.

---

### Author Response · Authors · 2022-11-16
**Comments on paper revision**

We would like to thank all the reviewers for their insightful comments and suggestions. We have answered all the questions under each reviewer's comments to address their concerns. We have also revised our paper accordingly (they have been put in blue bits):

1. We expanded the literature review in Section 2 of the revised paper to include more related works.

2. We added more experiments in Appendix E.5-E.8.

3. We relocated the discussion of 'Training Size' (in section 4.4 of the original paper) into Appendix E.4 of the revised paper (due to space limitations).

4. We added one sub-section 'Additional Experiments' in section 4.4 of the revised paper to reference all the added experiments in Appendix E.4-E.8.

---

### Decision · Program_Chairs · 2023-01-20

**Decision:**

Reject

**Justification For Why Not Higher Score:**

N/A

**Justification For Why Not Lower Score:**

N/A

**Metareview: Summary, Strengths And Weaknesses:**

This paper proposed a new time series generative model for bridging Temporal Convolutional Networks and Transformer via a layer-wise parallel structure. While the paper contains some interesting idea, reviewers raised major weakness concerns about weak empirical evaluations on toy data, limited novelty, and lack of convincing justification and insufficient comparisons with existing work. While the authors tried to respond to the review questions, some of major review concerns remain and the overall quality of this work is below the acceptance bar.